# Learning Generalized Label Distributions

## Abstract

Label ambiguity is pervasive in supervised learning, motivating a variety of representations beyond the traditional single-label setting. While label distribution (LD) provides a probabilistic description and has attracted increasing attention, we reveal its inherent limitations, including inconsistency with raw data, distortion of inter-sample order, and limited applicability. To address these issues, we introduce generalized label distribution (GLD), a unified representation that can perfectly recover raw data while preserving inter-sample order consistency, transform into existing forms of label representations without information loss, and capture out-of-distribution samples as well as negative label correlations. We further develop GLD learning algorithms and demonstrate their effectiveness through both theoretical analysis and extensive experiments.

## 1 Introduction

In supervised learning, the *single-label* (SL) setting has been the most widely adopted, where a sample is associated with a categorical variable. Growing recognition of label ambiguity has motivated the use of *logical label* (LL), i.e., a binary vector, to describe a sample, giving rise to multi-label learning (MLL) (Tsoumakas et al., 2010; Zhang & Zhou, 2013). In recent years, *label ranking* (LR) (Brinker et al., 2006; Lu & Jia, 2022) and *ternary label* (TL) (Lu & Jia, 2024) have also been introduced as alternative ways to model label ambiguity. Following this direction, *label distribution* (LD) has attracted increasing attention as a more fine-grained representation, using a probability distribution to describe a sample, leading to label distribution learning (LDL) (Geng, 2016).

However, in this paper, we argue that LD suffers from inherent limitations, including inconsistency with the raw data, disruption of inter-sample order, and limited applicability. These limitations not only challenge the claim that "*LDL represents a generalized form of MLL*" (Geng, 2016), but also expose critical shortcomings in current derivative tasks of LDL, e.g., label enhancement (LE) (Xu et al., 2019), joint LDL & LE (Liu et al., 2021; Jia et al., 2024), and classification-oriented LDL (Wang & Geng, 2019a; Wang et al., 2021), whose methodologies exhibit room for a more substantial refinement. Addressing these issues is both urgent and necessary, as they hinder further progress in learning with label ambiguity. In response, we propose a new form of representation, namely generalized label distribution (GLD), which extends LD to provide a more versatile and comprehensive characterization of label ambiguity. Specifically, GLD offers the following advantages:

- GLD serves as a truly unified form of label polysemy representation: it can be mapped upward to recover the raw data perfectly, and downward to derive other existing forms of label representations (Xu et al., 2020) (e.g., LD, LL, TL, LR, and SL) without any information loss.
- GLD can naturally characterize out-of-distribution (OOD) samples (Ren et al., 2019; Fort et al., 2021) and explicitly capture fine-grained negative correlations among labels (Xu et al., 2019).
- GLD does not counterintuitively distort the order relations across samples (Bergeron et al., 2008), making it more consistent with both machine learning models and human perception.

**Contributions & organizational structure** In this paper: (1) we conduct a thorough theoretical analysis of the inherent limitations of LD (Section 2); (2) we introduce GLD as a unified representation that addresses these limitations (Section 3.1); (3) we further propose several algorithms for effectively learning GLDs, which can also leverage and adapt existing LDL methods to extend their applicability to GLD (Section 3.2); (4) we provide both theoretical analyses and extensive experiments to demonstrate the effectiveness of the proposed GLD learning framework (Sections 3.3 and 4). The code will be available on Github soon, facilitating reproducibility and further research.

## 2 LIMITATIONS OF THE LABEL DISTRIBUTION

### 2.1 PRELIMINARIES

**Notation** Vectors are denoted by lowercase bold letters, e.g., $\boldsymbol{v}$, and the corresponding regular letter with subscript $i$, i.e., $v_i$, indicates its $i$-th element. Matrices are denoted by uppercase bold letters, e.g., $\boldsymbol{A}$, with $\boldsymbol{a}_i$ as the $i$-th *column* and $\boldsymbol{a}_{\bullet j}$ as the $j$-th *row*. $[k]$ is the set containing all integers from 1 up to $k$. $[\![\cdot]\!]$ is the Iverson bracket, which equals 1 if the condition is true and 0 otherwise. $\odot$ and $\oslash$ denote the element-wise multiplication and division, respectively.

**Assumption 2.1** (Projection-based label distribution). *Let $\boldsymbol{q} \in \prod_{j\in[c]}[a_j,\,b_j]_{\mathbb{R}}$ denote the raw data, where $c$ is the number of labels and $[a_j,\,b_j]_{\mathbb{R}}$ is the value range of each label $j$. The corresponding LD $\boldsymbol{d} \in \Delta^{c-1}$ is obtained via a projection operator $\mathrm{proj}(\cdot)$, i.e., $\boldsymbol{d} = \mathrm{proj}(\boldsymbol{q})$, where*

$$\Delta^{c-1} \triangleq \left\{ \boldsymbol{v} \in \mathbb{R}^c \,\middle|\, \mathbf{1}^\top \boldsymbol{v} = 1,\, \boldsymbol{v} \geq 0 \right\} \tag{1}$$

*is the $(c-1)$-dimensional probability simplex and $\mathbf{1}$ is a $c$-dimensional vector of all ones. Each $d_j$ of $\boldsymbol{d}$, namely description degree, indicates the degree to which the $j$-th label describes the sample.*

**Assumption 2.2** (Proportion-based label distribution). *Building on Assumption 2.1, the raw data $\boldsymbol{q}$ satisfies $\forall_{j\in[c]}(q_j \geq 0)$ and $\exists_{j\in[c]}(q_j > 0)$, then the LD is obtained by $\boldsymbol{d} = \boldsymbol{q}/\sum_{j\in[c]} q_j$.*

**Definition 2.3** (Logical label). *The LL is a binary vector $\boldsymbol{l} \in \{0,\,1\}^c$, where $l_j = 1$ indicates that the $j$-th label $y_j \in \mathcal{Y}$ is relevant to the sample, and $l_j = 0$ otherwise.*

### 2.2 LACK OF FIDELITY TO THE RAW DATA

In this subsection, we show that the LD representation is not strictly consistent with the raw data, which can lead to substantial information distortion. To illustrate this issue, we consider the conversion from an LD to its corresponding LL as an example (Kou et al., 2024). Specifically, we introduce the subset error rate based on the definition of subset accuracy (Zhang & Zhou, 2013). Given two LLs, $\boldsymbol{l}^*$ and $\boldsymbol{l}$, the subset error rate between them is defined as

$$\mathtt{S.Err.}(\boldsymbol{l}^*,\,\boldsymbol{l}) \triangleq [\![ \forall_{j\in[c]}((l_i^*)_j \neq (l_i)_j) ]\!]. \tag{2}$$

According to Kou et al., there exist two binarization algorithms to convert an LD to its corresponding LL, i.e., top-$k$-based and $\tau$-thresholding-based, denoted by $\mathrm{bin}_1(\cdot;\,k)$ and $\mathrm{bin}_2(\cdot;\,\tau)$, respectively, where $k \in [c]$ and $\tau \in [0,\,1]_{\mathbb{R}}$ are hyperparameters. The two algorithms can be formulated as follows:

$$\mathrm{bin}_1(\boldsymbol{d};\,k)_j \triangleq \left[\!\!\left[ y_j \in \arg\underset{y\in\mathcal{Y}}{\mathrm{top}\text{-}k}(\boldsymbol{d}) \right]\!\!\right], \tag{3}$$

$$\mathrm{bin}_2(\boldsymbol{d};\,\tau) \triangleq \mathrm{bin}_1(\boldsymbol{d};\,k^*), \tag{4}$$

where $k^* = \mathrm{card}(\min\{k \in [c] \,|\, \sum_{j\in[k]} \rho_j \geq \tau\})$, $\boldsymbol{\rho}$ is the sorted LD in *descending order*. For the top-$k$-based method, we have the following proposition.

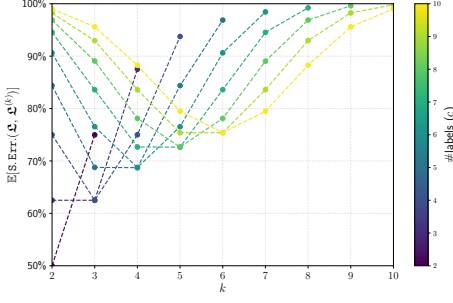

(a) $\mathbb{E}[\mathtt{S.Err.}(\boldsymbol{\mathfrak{L}}^*,\,\boldsymbol{\mathfrak{L}}^{(k)})]$ w.r.t. $k$ & $c$.

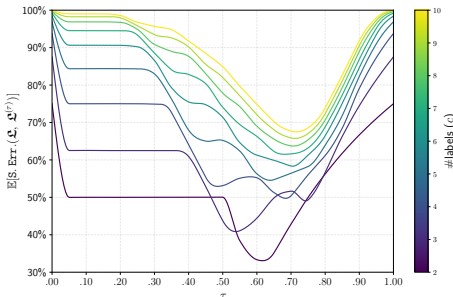

(b) $\mathbb{E}[\mathtt{S.Err.}(\boldsymbol{\mathfrak{L}}^*,\,\boldsymbol{\mathfrak{L}}^{(\tau)})]$ w.r.t. $\tau$ & $c$.

Figure 1: Visualization of the subset error rate between the LL derived from the raw data and that derived from the LD via two conversion methods: (a) top-$k$-based and (b) $\tau$-thresholding-based.

**Proposition 2.4.** *Let $\boldsymbol{\mathfrak{L}}^{(k)}$ denote the LL derived from the LD via the conversion rule in Equation (3); let $\boldsymbol{\mathfrak{L}}^*$ denote the LL directly derived from the raw data random vector $\boldsymbol{\mathfrak{P}}$ by thresholding at $\tau'$, i.e., $\mathfrak{L}_j = [\![ \mathfrak{P}_j \geq \tau' ]\!]$ for all $j \in [c]$. The raw data are independently drawn from a uniform distribution over $[a,\,b]_{\mathbb{R}}$, and the LD is obtained by its projection. Then, we have the following expectation:*

$$\mathbb{E}[\mathtt{S.Err.}(\boldsymbol{\mathfrak{L}}^*,\,\boldsymbol{\mathfrak{L}}^{(k)})] = 1 - \binom{c}{k}\left(\frac{b-\tau'}{b-a}\right)^k \left(\frac{\tau'-a}{b-a}\right)^{c-k} \geq 50\%. \tag{5}$$

*The equality holds if and only if $c = 2$ and $k = 1$.*

*Proof sketch.* $\mathfrak{L}^*$ arises as a Bernoulli vector determined by the uniform thresholding, while $\mathfrak{L}^{(k)}$ corresponds to selecting the $k$ largest order statistics, which directly yield the stated formula. $\qquad\square$

**Corollary 2.5.** *The following equation holds:*

$$\arg\min_{k} \; \mathbb{E}[\texttt{S.Err.}(\mathfrak{L}^*, \; \mathfrak{L}^{(k)})] = \left\lfloor \frac{c}{2} \right\rfloor. \tag{6}$$

Let $\mathfrak{L}^{(\tau)}$ denote the LL derived from the LD by Equation (4). For the $\tau$-thresholding-based method, obtaining a closed-form expression of the expectation of the error rate is intractable. Therefore, we resort to Monte Carlo estimation under the similar setting in Proposition 2.4, with a sampling size of $10^6$ and varying the number of labels $c \in [2, 10]_{\mathbb{Z}}$. The results are visualized in Figure 1b, where we also include the visualization for the top-$k$-based method for comparison, as shown in Figure 1a.

*Remark* 2.6. As established in Proposition 2.4 and corroborated by the empirical evidence in Figure 1, it is observed that the top-$k$-based method consistently incurs an error rate above $50\%$, even when using the recommended choice of $k$, i.e., $\lfloor c/2 \rfloor$ specified in Corollary 2.5; the $\tau$-thresholding-based method achieves comparatively lower error rates, yet they remain exceeding $30\%$, with $\tau$ optimally selected within the interval $[0.5, 0.75]_{\mathbb{R}}$. For both methods, the minimum attainable error rate grows unfavorably with the number of labels $c$.

## 2.3 DISRUPTION OF INTER-SAMPLE ORDER

In this subsection, we show that LD can counterintuitively distort the order relations across samples, which is undesirable in many scenarios. To quantify this effect, we adopt Kendall's tau, a widely used metric to measure the ordinal association between two sequences. Its type-A definition is as follows:

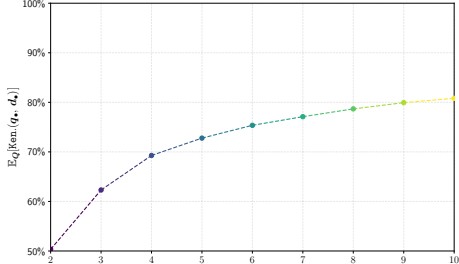

Figure 2: Visualization of Monte Carlo estimation of inter-sample order consistency, i.e., $\mathbb{E}_{\boldsymbol{Q}}[\texttt{Ken.}(\boldsymbol{q}_\bullet, \boldsymbol{d}_\bullet)]$ w.r.t. $c$.

$$\texttt{Ken.}(\boldsymbol{u}, \boldsymbol{v}) \triangleq \frac{2\sum_{i<j} \text{sign}(u_i - u_j)\,\text{sign}(v_i - v_j)}{n(n-1)}. \tag{7}$$

**Proposition 2.7.** *Assume that the raw data matrix $\boldsymbol{Q} \in [0, 1]_{\mathbb{R}}^{c \times n}$ has entries drawn independently from the uniform distribution. The corresponding LD matrix is obtained by $\boldsymbol{D} = \text{proj}(\boldsymbol{Q})$. Treating each row as a sequence across samples, the inter-sample order consistency can be quantified by*

$$\begin{aligned}
\mathbb{E}_{\boldsymbol{Q}}[\texttt{Ken.}(\boldsymbol{q}_\bullet, \boldsymbol{d}_\bullet)] &= \mathbb{E}_{\boldsymbol{Q}}[\text{sign}((\boldsymbol{q}_\bullet - \boldsymbol{q}_\bullet')^\top(\boldsymbol{q}_\bullet \oslash \boldsymbol{s} - \boldsymbol{q}_\bullet' \oslash \boldsymbol{s}))] \\
&= 2\mathbb{P}((\boldsymbol{q}_\bullet - \boldsymbol{q}_\bullet')^\top(\boldsymbol{q}_\bullet \oslash \boldsymbol{s} - \boldsymbol{q}_\bullet' \oslash \boldsymbol{s}) > 0) - 1,
\end{aligned} \tag{8}$$

*where $\boldsymbol{q}_\bullet$ and $\boldsymbol{q}_\bullet'$ are two independent sequences drawn from the same distribution as the rows of $\boldsymbol{Q}$, and $s_i = \sum_{j \in [c]} (\boldsymbol{q}_i)_j$ is the LD normalization factor for each sample $i$.*

Proposition 2.7 follows directly since the $\text{sign}(\cdot)$ function always yields $\pm 1$ in the absence of ties. For the special case $c = 2$, the probability term equals $3/4$ and the expected Kendall's tau is $0.5$. For $c > 2$, however, the expectation is analytically intractable due to the high-dimensional integrals involved. Therefore, we employ Monte Carlo estimation, fixing the number of samples at $10^6$ and varying the number of labels $c \in [2, 10]_{\mathbb{Z}}$. The results are visualized in Figure 2.

*Remark* 2.8. As demonstrated in Proposition 2.7 and Figure 2, the expected Kendall's tau increases with the number of labels $c$, yet it remains significantly below 1, indicating that LD can substantially distort the inter-sample order. Note that this apparent increase with $c$ is primarily due to the uniform distribution spreading the values more evenly across labels, which reduces the likelihood of inconsistencies on average; values across different $c$ may not be directly comparable.

## 2.4 LIMITED PRACTICAL APPLICABILITY

In this subsection, we simply discuss the flaws of LD in practical applications using some examples.

*Remark* 2.9. Under Assumption 2.1, the practical limitations of LD can be summarized as follows:

- Inability to reconstruct the raw data: LD can *not* recover the original information, such as absolute magnitude and label annotation (Xu et al., 2020). For example, different colors (e.g., Magenta $\boldsymbol{q} = \langle 255, 0, 255 \rangle^\top$ and Patriarch $\boldsymbol{q}' = \langle 128, 0, 128 \rangle^\top$ in the RGB color space) may share the same LD ($\boldsymbol{d} = \langle 0.5, 0, 0.5 \rangle^\top$) in an LD color space.
- Lack of mechanism to handle OOD data: LD assumes that every label is relevant to the sample to some extent, which may *not* hold in real-world scenarios (Wu et al., 2025). For example, in a emotion recognition task, an image (e.g., a neutral face) may not convey any of the predefined emotions (e.g., the primary emotions like happiness, sadness, anger, fear, surprise, and disgust), but LD would still assign non-zero description degrees to all emotions.

Given the aforementioned limitations of LD, a more expressive and flexible representation is needed. This motivates the concept of generalized label distribution, which we present in the next section.

## 3 GENERALIZED LABEL DISTRIBUTION

### 3.1 FORMULATION OF GENERALIZED LABEL DISTRIBUTION

*Remark* 3.1. Assume a GLD mapping $\eta(\cdot)$. To address the limitations inherent in traditional LD, a GLD should satisfy the following properties:

- Regarding Remark 2.6, GLD should allow conversion to other forms of label representations, e.g., LD, LL, TL, LR, and SL, without any information loss. For example, in the case of LL, there should exist a mapping function GLD2LL($\cdot$) such that, for any LL $\boldsymbol{l}^*$ derived from the raw data $\boldsymbol{q}$, the equation $\texttt{S.Err.}(\boldsymbol{l}^*, \text{GLD2LL}(\eta(\boldsymbol{q}))) = 0$ holds.
- Regarding Remark 2.8, GLD should preserve the inter-sample order, i.e., for the $j$-th *row* of any raw data matrix $\boldsymbol{Q}$, the equation $\texttt{Ken.}(\boldsymbol{q}_{\bullet j}, \eta(\boldsymbol{Q})_{\bullet j}) = 1$ holds.
- Regarding Remark 2.9, GLD should be capable of handling OOD data while providing a clear characterization of both positive and negative correlations; $\eta(\cdot)$ should be bijective to ensure that the raw data can be accurately reconstructed, i.e., $\boldsymbol{q} = \eta^{-1}(\eta(\boldsymbol{q}))$ for any raw data $\boldsymbol{q}$.

With these properties in mind, we present the formal definition of GLD as follows:

**Definition 3.2** (Generalized label distribution). Given the raw data $\boldsymbol{q} \in \prod_{j \in [c]} [a_j, b_j]_{\mathbb{R}}$, the GLD is defined as
$$[-1, 1]_{\mathbb{R}}^c \ni \boldsymbol{g} = \eta(\boldsymbol{q}; \boldsymbol{a}, \boldsymbol{b}) \triangleq 2(\boldsymbol{q} - \boldsymbol{a}) \oslash (\boldsymbol{b} - \boldsymbol{a}) - 1. \tag{9}$$
Conversely, the raw data can be recovered from the GLD by
$$\prod_{j \in [c]} [a_j, b_j]_{\mathbb{R}} \ni \boldsymbol{q} = \eta^{-1}(\boldsymbol{g}; \boldsymbol{a}, \boldsymbol{b}) \triangleq \frac{(\boldsymbol{g} + 1) \odot (\boldsymbol{b} - \boldsymbol{a})}{2} + \boldsymbol{a}. \tag{10}$$

Based on Definition 3.2, we have the following theorem regarding the expressiveness of GLD.

**Theorem 3.3.** *Let $\mathfrak{P} \sim \mathcal{Q}$ be the random vector of the raw data, $\mathfrak{D} = \text{proj}(\mathfrak{P})$, and $\mathfrak{G} = \eta(\mathfrak{P})$. Under Assumption 2.1, we have*
$$I(\mathfrak{P}; \mathfrak{D}) \leq I(\mathfrak{P}; \mathfrak{G}), \tag{11}$$
*where $I(\cdot; \cdot)$ is the mutual information. The equality holds if $\mathcal{Q} \subseteq \psi(\Delta^{c-1})$, where $\psi(\cdot)$ is an invertible transformation and $\mathcal{Q}$ is the distribution of the raw data.*

*Proof.* See Appendix A.1 for details. □

In practical, the mapping $\psi(\Delta^{c-1})$ can be a scaled simplex $\kappa\Delta^{c-1}$ ($\kappa > 0$), a.k.a. the *Aitchison simplex* (Aitchison, 1994), i.e., the raw data are compositional data constrained to a fixed sum $\kappa$. It should be noted that this is a rather stringent condition, rarely satisfied in general scenarios. Theorem 3.3 indicates that GLD provides a more expressive and faithful representation compared to conventional LD, supporting tasks that rely on nuanced label correlations.

**Underlying philosophy** The essence of GLD lies in the concept of *net probability* from a statistical perspective. It reflects the intrinsic association between samples and labels, analogous to *odds*, but within a more interpretable range that can express both positive and negative correlations. Whereas previous work in LD focused on the description degrees, i.e., answering "*To what extent does a label describe a sample?*", GLD acts as a set of correlation coefficients, i.e., answering "*To what extent is a sample associated with a label?*". In this sense, we name the values in GLD as *relative degrees*. Table 1 illustrates the conversion capabilities among various label ambiguity representations, showing that GLD can be mapped to all other forms seamlessly and losslessly. Additional insights of Table 1 are provided in Appendix B.

Table 1: Conversion capabilities among label ambiguity representations. ✓ indicates that the conversion can be performed losslessly.

| From | To | | | | | |
|------|-----|-----|-----|-----|-----|-----|
|      | GLD | LD | TL | LL | LR | SL |
| GLD  |     | ✓  | ✓  | ✓  | ✓  | ✓  |
| LD   |     |    |    |    | ✓  | ✓  |
| TL   |     |    |    |    |    |    |
| LL   |     |    |    |    |    |    |
| LR   |     |    |    |    |    | ✓  |
| SL   |     |    |    |    |    |    |

Given the numerous benefits of GLD, one may wonder how to design a learning framework capable of effectively learning GLDs. This is the focus of the next subsection.

## 3.2 LEARNING GENERALIZED LABEL DISTRIBUTIONS

**Problem formulation** Let $\mathcal{X} = \mathbb{R}^m$ denote the input space, and $\mathcal{G} \subseteq [-1, 1]_{\mathbb{R}}^c$ the output space. Given the training set $\mathcal{S} = \{(\boldsymbol{x}_i, \boldsymbol{g}_i)\}_{i\in[n]}$, the goal is to find a mapping $f : \mathcal{X} \mapsto \mathcal{G}$, where $\boldsymbol{x}_i \in \mathcal{X}$ and $\boldsymbol{g}_i \in \mathcal{G}$ for all $i \in [n]$. The function $f$ is not only required to accurately predict the GLD $\boldsymbol{g}$ for any unseen instance $\boldsymbol{x}$, but also to perform well on other paradigms, e.g., LDL, MLL, etc.

Inspired by Geng, we discuss three strategies to learn GLDs, i.e., problem transformation, algorithm adaptation, and specialized algorithm. Note that these following methods are designed to facilitate fair comparisons with their respective reference models.

### 3.2.1 PROBLEM TRANSFORMATION & ALGORITHM ADAPTATION

**Problem transformation** The problem of learning GLDs can be reformulated as a series of constrained regression tasks, which bears similarity to classical methods in LDL and multi-target regression (Liu et al., 2009; Geng & Hou, 2015). Like them, we employ the support vector regression (SVR) algorithm as the underlying model. Note that we do *not* adopt the same problem transformation strategy used in (Geng, 2016), as it would compromise the intrinsic structure of the target data. To address the bounded nature of GLDs, we incorporate a forward-regressor-inverse transformation framework. Specifically, before training, the target values are first mapped to the real line via $\mathrm{arctanh}(\mathrm{clip}(\cdot; -1+\varepsilon, 1-\varepsilon))$, where $\mathrm{clip}(\cdot; a, b)$ restricts the input to the specified range $[a, b]_{\mathbb{R}}$, and $\varepsilon$ is a small positive constant to avoid numerical issues. After training, the predictions are mapped back to the original space using $\tanh(\cdot)$. The resulting method, referred to as GLD-SVR, is straightforward and intuitive.

**Algorithm adaptation** We can also naturally adapt some existing algorithms to deal with GLDs, among which we consider the $k$-nearest neighbor algorithm as an example, since there are already some extended variants for LDL and MLL (Zhang & Zhou, 2007; Geng, 2016). For a given test sample $\boldsymbol{x}$, we first determine the set $\mathcal{N}(\boldsymbol{x})$ of its $k$ nearest neighbors from the training set. The GLD prediction is then obtained by averaging the GLDs of these neighbors, i.e., $1/k \sum_{\boldsymbol{x}_i \in \mathcal{N}(\boldsymbol{x})} \boldsymbol{g}_i$. The resulting method, referred to as GLD-$k$NN, is also simple yet effective.

### 3.2.2 SPECIALIZED ALGORITHM

Different from the above two strategies, we can also design specialized algorithms to directly match the GLD learning problem. Here, we consider a simple network-based model with a single linear layer followed by a $\tanh(\cdot)$ activation function, i.e., $f(\boldsymbol{x}) = \tanh(\boldsymbol{W}\boldsymbol{x})$, where $\boldsymbol{W} \in \mathbb{R}^{c \times m}$ is the weight matrix to be learned.

*Remark* 3.4. Consider the GLD model where each observation is assumed to be corrupted by multivariate Gaussian noise, i.e., $\boldsymbol{g}_i = f(\boldsymbol{x}_i) + \boldsymbol{\varepsilon}_i$ with $\varepsilon_i \sim \text{Gaussian}(\boldsymbol{0}, \boldsymbol{\Sigma})$ for all $i \in [n]$, where $\boldsymbol{\Sigma} \in \mathbb{R}^{c \times c}$ is the covariance matrix.

Let $\varphi(\boldsymbol{x}, \boldsymbol{g}; \boldsymbol{\Sigma})$ denote the probability density function, the corresponding negative log-likelihood can be expressed as

$$-\log \prod_{i \in [n]} \varphi(\boldsymbol{x}_i, \boldsymbol{g}_i; \boldsymbol{\Sigma}) = \underbrace{\frac{nc}{2} \log(2\pi) + \frac{n}{2} \log \det(\boldsymbol{\Sigma})}_{\text{constant}} + \frac{1}{2} \sum_{i \in [n]} \underbrace{(\boldsymbol{g}_i - f(\boldsymbol{x}_i))^\top \boldsymbol{\Sigma}^{-1} (\boldsymbol{g}_i - f(\boldsymbol{x}_i))}_{\text{squared Mahalanobis distance}}.$$
(12)

Equation (12) reveals that, apart from an additive constant, the negative log-likelihood naturally reduces to the sum of squared Mahalanobis distances between the predicted outputs and the observed GLDs. Consequently, adopting the Mahalanobis distance as a loss function is theoretically well justified. Formally, the loss function can be defined as follows:

$$\ell(\boldsymbol{W}; \{(\boldsymbol{x}_i, \boldsymbol{g}_i)\}_{i \in [n]}) = \frac{1}{n} \sum_{i \in [n]} (\boldsymbol{g}_i - \tanh(\boldsymbol{W}\boldsymbol{x}_i))^\top \boldsymbol{\Sigma}^{-1} (\boldsymbol{g}_i - \tanh(\boldsymbol{W}\boldsymbol{x}_i)).$$
(13)

**Optimization** We can learn the optimal model parameters $\boldsymbol{W}^*$ by minimizing the empirical loss, i.e., $\boldsymbol{W}^* = \arg\min_{\boldsymbol{W}}(\ell)$, which can be solved using gradient-based optimization methods.

*Remark* 3.5. $\ell$ is differentiable and its gradient w.r.t. $\boldsymbol{W}$ is given by

$$\frac{\partial \ell(\boldsymbol{W})}{\partial \boldsymbol{W}} = -\frac{2}{n} \sum_{i \in [n]} ((\boldsymbol{\Sigma}^{-1}(\boldsymbol{g}_i - \tanh(\boldsymbol{W}\boldsymbol{x}_i))) \odot \text{sech}^2(\boldsymbol{W}\boldsymbol{x}_i))\boldsymbol{x}_i^\top,$$
(14)

derivation of which is provided in Appendix A.2, where $\text{sech}(\cdot)$ is the hyperbolic secant function.

In practice, $\boldsymbol{\Sigma}$ needs to be updated iteratively. However, in the early stages of training, the residuals are not yet reliable, hence retaining $\boldsymbol{I}$ during the first half of the training process is advisable. Let $t'$ denote the current iteration and $t$ the maximum number of iterations. In the latter half of the training, we estimate $\boldsymbol{\Sigma}'$ using the Ledoit-Wolf method (Ledoit & Wolf, 2004) when $t' \equiv 0 \pmod{r}$, where $r$ is a predefined frequency. The inverse covariance matrix is then computed as $\alpha(\boldsymbol{\Sigma}')^{-1} + (1 - \alpha)\boldsymbol{I}$, where $\alpha = t'/t$ is an annealing factor. This helps avoid abrupt gradient fluctuations.

The resulting method is referred to as GLD-BFGS when employing the L-BFGS-B algorithm (Zhu et al., 1997) for optimization, which is a quasi-Newton method that approximates the Hessian matrix using gradient evaluations. Importantly, this specialized algorithm strategy is not only limited to GLD-BFGS itself, but also can be integrated into and adapted from existing LDL methods, extending their capability to support GLD learning. Further details are provided in Section 4.1.

**Time complexity analysis** Let $\boldsymbol{X}$ denote the feature matrix, $\boldsymbol{G}$ the GLD matrix. The overall time cost of GLD-BFGS is primarily influenced by the following calculations: the forward pass step $\tanh(\boldsymbol{W}\boldsymbol{X})$ requires $\mathcal{O}(nmc)$; the squared Mahalanobis distance computation in Equation (13), i.e., $(\boldsymbol{G} - \tanh(\boldsymbol{W}\boldsymbol{X})) \odot \boldsymbol{\Sigma}^{-1}(\boldsymbol{G} - \tanh(\boldsymbol{W}\boldsymbol{X}))$, has a complexity of $\mathcal{O}(nc^2)$; the gradient calculation in Equation (14), vectorized as $\boldsymbol{\Sigma}^{-1}((\boldsymbol{G} - \tanh(\boldsymbol{W}\boldsymbol{X})) \odot (\boldsymbol{1} - \tanh^2(\boldsymbol{W}\boldsymbol{X})))\boldsymbol{X}^\top$, involves a complexity of $\mathcal{O}(npc + nc^2)$. The calculation of $\boldsymbol{\Sigma}^{-1}$ has a complexity of $\mathcal{O}(nc^2)$, but this is performed infrequently. Therefore, the overall time complexity of GLD-BFGS is $\mathcal{O}(tnc^2 + tnpc)$.

### 3.3 THEORETICAL ANALYSIS

**Theorem 3.6.** *Let $\mathfrak{D}^*$ denote the underlying LD derived from the raw data random vector under Assumption 2.2; let $\mathfrak{D}'$ and $\mathfrak{D}$ be the predictions of GLD-kNN and LD-kNN, respectively. Suppose both $\mathfrak{D}$ and $\mathfrak{D}'$ are unbiased estimators of $\mathfrak{D}^*$. Then, the variance of $\mathfrak{D}'$ with respect to $\mathfrak{D}^*$ is no larger than that of $\mathfrak{D}$, i.e., $\mathbb{V}[\mathfrak{D}'_j - \mathfrak{D}^*_j] \leq \mathbb{V}[\mathfrak{D}_j - \mathfrak{D}^*_j]$ for all $j \in [c]$. The equality holds under the same condition as that required for the equality case in Theorem 3.3.*

**Theorem 3.7.** *Let $\mathfrak{L}^*$ denote the underlying LL derived from the raw data random vector $\mathfrak{P}$ by thresholding at $\boldsymbol{\tau}'$; let $\mathfrak{L}'$ and $\mathfrak{L}$ be the predictions of GLD-kNN and LL-kNN, respectively. Suppose*

both $\mathfrak{L}$ and $\mathfrak{L}'$ are unbiased estimators of $\mathfrak{L}^*$. Then, for the $j$-th label, $\mathbb{V}[\mathfrak{L}'_j - \mathfrak{L}^*_j] \leq \mathbb{V}[\mathfrak{L}_j - \mathfrak{L}^*_j]$ holds when $\sqrt{\mathbb{V}[\mathfrak{P}_j]} \leq 2|\tau'_j|\sqrt{\mathbb{P}(\mathfrak{P}_j \geq \tau'_j)(1 - \mathbb{P}(\mathfrak{P}_j \geq \tau'_j))}$ holds.

**Theorem 3.8.** *Let $\hat{\mathfrak{R}}_n$ be the empirical Rademacher complexity w.r.t. $\mathcal{S}$ with $n$ samples; $\mathcal{H}$ be a family of functions; and $\mathcal{H}_j$ be the $j$-th component of the output of $\mathcal{H}$ for all $j \in [c]$. For the squared Mahalanobis distance loss function $\mathrm{M}^2(\cdot, \cdot)$ with covariance matrix $\mathbf{\Sigma}$, we have*

$$\hat{\mathfrak{R}}_n(\mathrm{M}^2 \circ \tanh \circ \mathcal{H} \circ \mathcal{S}) \leq 4\sqrt{2c}\|\mathbf{\Sigma}^{-1}\| \sum_{j \in [c]} \hat{\mathfrak{R}}_n(\mathcal{H}_j \circ \mathcal{S}). \tag{15}$$

**Theorem 3.9.** *Define a family of functions $\mathcal{H}$ and the $j$-th component of the output $\mathcal{H}_j = \{\boldsymbol{x} \mapsto \boldsymbol{w}_j \cdot \boldsymbol{x}\}$, where $\|\boldsymbol{w}_j\|_2 \leq \xi$ for all $j \in [c]$. Let $\mathcal{F}$ be the family of functions for methods corresponding to Section 3.2.2, for any $\delta > 0$, with probability at least $1 - \delta$, for all $f \in \mathcal{F}$, we have*

$$\mathbb{E}_{\boldsymbol{x} \sim \mathcal{D}} \left[ \mathrm{M}^2(\boldsymbol{g}_{\boldsymbol{x}}, f(\boldsymbol{x})) \right] \leq \frac{1}{n} \sum_{i \in [n]} \mathrm{M}^2(\boldsymbol{g}_i, f(\boldsymbol{x}_i)) + 12c\|\mathbf{\Sigma}^{-1}\|\sqrt{\frac{\log(2/\delta)}{2n}} +$$
$$\frac{8\xi c\sqrt{2c}\|\mathbf{\Sigma}^{-1}\|}{\sqrt{n}} \max_{i \in [n]} \|\boldsymbol{x}_i\|, \tag{16}$$

*where $\mathcal{D}$ is the underlying distribution of the input space, and $\boldsymbol{g}_{\boldsymbol{x}}$ is the ground-truth GLD of $\boldsymbol{x}$.*

*Proof.* Proofs of these above theorems are provided in Appendices A.3 to A.6, respectively. $\square$

# 4 EXPERIMENTS

## 4.1 EXPERIMENTAL SETUP

**Datasets** We construct an artificial dataset `artf` similar to (Geng, 2016), with only a limited number of training samples to simulate a challenging learning scenario; details are given in Appendix C. The real-world datasets have been examined in previous research (Spyromitros-Xioufis et al., 2016; González et al., 2021), including: a facial emotion recognition dataset of Japanese female faces, denoted as `jaf`; a geochemical composition dataset collected from the topsoil of the Swiss Jura region, denoted as `jura`; a water quality dataset `wq`; two datasets for price prediction, `atp` and `scm`; and a dataset on heating/cooling loads of buildings (i.e., energy efficiency), denoted as `enb`.

**Evaluation metrics** We do *not* directly evaluate the GLD prediction task itself, as its practical application is unexplored. Instead, we assess performance through metrics in these following versatile tasks, where $\downarrow$ ($\uparrow$) indicates the smaller (larger) the better; see Appendix C for more details.

- OOD classification: For GLD, if all relative degree values are negative, the sample is regarded as an OOD instance. While LD cannot provide explicit OOD information, the situation where all labels are related to a sample is uncommon. Therefore, by leveraging the uniform distribution, we design a new metric `OOD Err.` $\downarrow$ to make a relatively fair comparison.
- Intra/inter-sample ranking: The Spearman rank correlation coefficient `Spear.` $\uparrow$ is commonly used for evaluating label ranking performance (Jia et al., 2023). In addition, we compute Kendall's tau for each *row* of the predicted/ground-truth matrix to assess the consistency of inter-sample ordering, and denote this metric as `Ken.'` $\uparrow$.
- LL prediction: We convert the model outputs into LLs and evaluate them using the Hamming distance `Ham.` $\downarrow$ and subset accuracy `S. Acc.` $\uparrow$ (Zhang & Zhou, 2013). For LD, the binarization algorithm $\mathrm{bin}_2$ is adopted.
- LD prediction: We evaluate using the Clark distance `Clark` $\downarrow$ (Geng, 2016) and a "regularized" K-L divergence (KLD) denoted as $\mu_{\mathrm{KLD}}$ $\uparrow$ (Li et al., 2025).

**Baselines** GLD-SVR, GLD-$k$NN, and GLD-BFGS in Section 3.2 correspond to baselines proposed in (Geng, 2016), which we denote as LD-SVR, LD-$k$NN, and LD-BFGS for clarity. In addition, we design the following GLD methods based on the specialized algorithm strategy:

- GLD-DF: This method is based on an ensemble method, denoted as LD-DF, which assigns different data batches to different label pairs (González et al., 2021). In our adaptation, we replace its base estimators with GLD-BFGS models, and substitute the built-in LD-$k$NN with GLD-$k$NN.

Table 2: Experimental results on datasets `artf`, `jura`, and `wq` formatted as (`mean ± std`).

| Algorithms | Clark ↓ | $\mu_{KLD}$ ↑ | Ham. ↓ | S. Acc. ↑ | Spear. ↑ | Ken.′ ↑ | OOD Err. ↓ | Avg. Rank |
|---|---|---|---|---|---|---|---|---|
| | | | | `artf` | | | | |
| LD-SVR | .0307±.005 | 98.58%±.005 | ● .2338±.046 | ● 40.85%±.097 | ● .9640±.028 | ● .5083±.065 | ● 12.70%±.076 | 6.14 |
| GLD-SVR | .0307±.006 | 98.57%±.005 | .0143±.014 | 95.70%±.041 | .9763±.022 | .9502±.014 | 03.35%±.041 | 2.86 |
| LD-kNN | .0475±.011 | 96.40%±.011 | ● .2340±.047 | ● 40.65%±.102 | .9333±.036 | ● .4942±.064 | ● 13.95%±.071 | 9.43 |
| GLD-kNN | .0476±.008 | 96.41%±.011 | .0547±.031 | 84.10%±.088 | .9328±.037 | .9046±.024 | 08.25%±.056 | 6.86 |
| LD-BFGS | ● .0783±.010 | ● 90.90%±.022 | ● .2532±.048 | ● 36.85%±.094 | ● .9470±.036 | ● .5063±.065 | ● 09.30%±.063 | 10.3 |
| GLD-BFGS | .0444±.009 | 96.55%±.010 | .0670±.030 | 80.40%±.088 | .9565±.030 | .9589±.014 | 04.25%±.042 | 5.29 |
| LD-DF | ● .0492±.007 | ● 96.38%±.010 | ● .2497±.047 | ● 37.10%±.096 | ● .9200±.046 | ● .5103±.066 | ● 16.35%±.082 | 10.1 |
| GLD-DF | **.0237**±.004 | **99.05%**±.003 | **.0118**±.013 | **96.45%**±.038 | .9758±.021 | **.9648**±.011 | **02.45%**±.031 | **1.14** |
| LD-LRR | ● .0783±.010 | ● 90.90%±.022 | ● .2532±.048 | ● 36.85%±.094 | ● .9470±.036 | ● .5062±.065 | ● 09.30%±.063 | 10.4 |
| GLD-LRR | .0444±.009 | 96.55%±.010 | .0670±.030 | 80.40%±.088 | .9565±.030 | .9589±.014 | 04.25%±.042 | 5.14 |
| LD-Delta | .0285±.012 | 98.41%±.014 | ● .2413±.048 | ● 38.95%±.098 | ○ .9683±.027 | ● .5103±.066 | ● 10.75%±.069 | 6.43 |
| GLD-Delta | .0299±.006 | 98.56%±.006 | .0303±.020 | 90.90%±.059 | .9590±.027 | .9559±.014 | 06.60%±.058 | 3.86 |
| | | | | `jura` | | | | |
| LD-SVR | .2709±.027 | ● 91.90%±.018 | ● .4687±.033 | ● 02.23%±.023 | .9217±.030 | ● .3160±.053 | ● 61.78%±.090 | 9.29 |
| GLD-SVR | .2665±.025 | 92.55%±.016 | .1137±.027 | 68.27%±.069 | .9235±.030 | .5419±.063 | 17.16%±.059 | 5.71 |
| LD-kNN | .2833±.027 | ○ 91.29%±.019 | ● .4752±.036 | ● 03.26%±.025 | .9177±.031 | ● .2806±.050 | ● 61.44%±.091 | 10.4 |
| GLD-kNN | .2768±.029 | 90.39%±.024 | .1184±.029 | 68.04%±.073 | .9188±.031 | .4981±.060 | 20.61%±.059 | 8.00 |
| LD-BFGS | ● .3009±.031 | 92.60%±.015 | ● .4803±.033 | ● 03.12%±.026 | .9348±.029 | ● .3120±.050 | ● 57.12%±.097 | 8.71 |
| GLD-BFGS | .2530±.024 | 92.86%±.016 | .1155±.027 | 68.38%±.070 | .9301±.030 | .5772±.056 | 15.43%±.052 | 4.29 |
| LD-DF | ● .2562±.025 | 93.56%±.014 | ● .4795±.036 | ● 03.26%±.029 | .9308±.028 | ● .3149±.050 | ● 59.05%±.095 | 7.43 |
| GLD-DF | **.2394**±.023 | 93.67%±.014 | .1151±.027 | 69.42%±.068 | .9291±.030 | **.6028**±.051 | 15.91%±.052 | 2.86 |
| LD-LRR | .2595±.025 | ○ 93.59%±.014 | ● .4792±.034 | ● 02.93%±.027 | .9336±.031 | ● .3125±.050 | ● 59.21%±.097 | 7.57 |
| GLD-LRR | .2733±.099 | 90.21%±.129 | .1130±.026 | 68.91%±.070 | .9189±.070 | .5553±.099 | **15.38%**±.051 | 5.86 |
| LD-Delta | .2494±.024 | ○ **94.33%**±.014 | ● .4720±.037 | ● 03.71%±.031 | ○ **.9403**±.030 | ● .3264±.052 | ● 58.96%±.093 | 5.00 |
| GLD-Delta | .2483±.022 | 93.42%±.017 | **.1102**±.027 | **70.55%**±.068 | .9319±.030 | .5950±.054 | 18.08%±.057 | 2.86 |
| | | | | `wq` | | | | |
| LD-SVR | 3.4504±.030 | ● 00.48%±.006 | ● .1844±.009 | ● 05.51%±.021 | ● .3400±.021 | ● .0809±.024 | ● 78.89%±.036 | 10.4 |
| GLD-SVR | 3.6075±.010 | 00.77%±.006 | .1649±.010 | 11.50%±.032 | .3643±.021 | .1110±.022 | 70.23%±.046 | 8.00 |
| LD-kNN | ● 2.7350±.038 | 20.28%±.025 | ● .2317±.009 | ● 02.46%±.015 | .3628±.023 | ○ .2388±.022 | ○ 75.84%±.036 | 7.57 |
| GLD-kNN | **2.7234**±.038 | 20.55%±.024 | .1644±.010 | 11.02%±.031 | .3631±.026 | .2306±.023 | 69.30%±.043 | 5.71 |
| LD-BFGS | ○ 3.1244±.032 | ○ 27.57%±.014 | ● .2646±.011 | ● 01.20%±.010 | ○ .3901±.022 | ● .2516±.019 | ● 77.31%±.039 | 7.57 |
| GLD-BFGS | 3.1396±.037 | 25.83%±.034 | .1640±.010 | 10.70%±.030 | .3706±.032 | .2142±.030 | 68.62%±.047 | 6.57 |
| LD-DF | ○ 3.1184±.032 | 29.36%±.014 | ● .2556±.009 | ● 01.58%±.011 | ○ .4092±.022 | ○ .2730±.019 | ● 77.18%±.040 | 5.43 |
| GLD-DF | 3.1278±.031 | 29.07%±.015 | .1623±.010 | 10.99%±.029 | .3985±.023 | .2508±.018 | 68.02%±.048 | 4.29 |
| LD-LRR | 3.1235±.032 | ○ 27.41%±.014 | ● .2679±.010 | ● 01.22%±.011 | ○ .3893±.022 | .2469±.020 | ● 77.68%±.038 | 8.14 |
| GLD-LRR | 3.1272±.033 | 26.81%±.014 | .1650±.010 | 10.61%±.028 | .3679±.023 | .2183±.018 | 68.76%±.045 | 6.71 |
| LD-Delta | 3.1277±.031 | ○ **30.24%**±.017 | ● .2396±.011 | ● 02.23%±.012 | ● **.4162**±.022 | ○ **.2778**±.020 | ● 76.26%±.041 | 5.29 |
| GLD-Delta | 3.1221±.032 | 29.10%±.016 | **.1607**±.010 | 11.87%±.031 | .4025±.024 | .2540±.019 | **66.33%**±.044 | 2.29 |

- GLD-LRR: The reference method LD-LRR designs a loss function that incorporates label ranking relationships (Jia et al., 2023). We convert the predicted GLD into LD to make it compatible with this loss, and replace the original KLD with the squared Mahalanobis distance.
- GLD-Delta: The corresponding method, denoted as LD-Delta, optimizes a "regularized" KLD to ensure that the majority of samples are approximately predicted, thereby mitigating strong interference from outlier data (Li et al., 2025). To adapt it for GLD, both the loss and the worst-case expectation are computed using the squared Mahalanobis distance.

**Methodology** To ensure a fair comparison, for each dataset and for each method we conduct ten-fold experiments repeated 10 times, and the average performance is recorded. Representative results are in Table 2. The best and second-best results are highlighted in **bold** and underline, respectively. ● (○) indicates "GLD-X is statistically superior (inferior) to the comparing methods LD-X" (pairwise $t$-test at 0.05 significance level); if neither ● nor ○ is present, there is no significant difference.

## 4.2 DISCUSSION

**The artificial dataset** The first observation is that, across almost all evaluation metrics, the proposed GLD-based methods demonstrate clear advantages over their LD counterparts. In particular, the improvements are most pronounced in LL prediction, inter-sample ranking, and OOD classification.[1] These three aspects correspond directly to the limitations of LD discussed in Remarks 2.6,

---

[1]It should be noted that the observed performance gap primarily stems from the representational limitations of LD itself. The conversion strategies (e.g., $bin_2$) have already been applied in the most effective way possible.

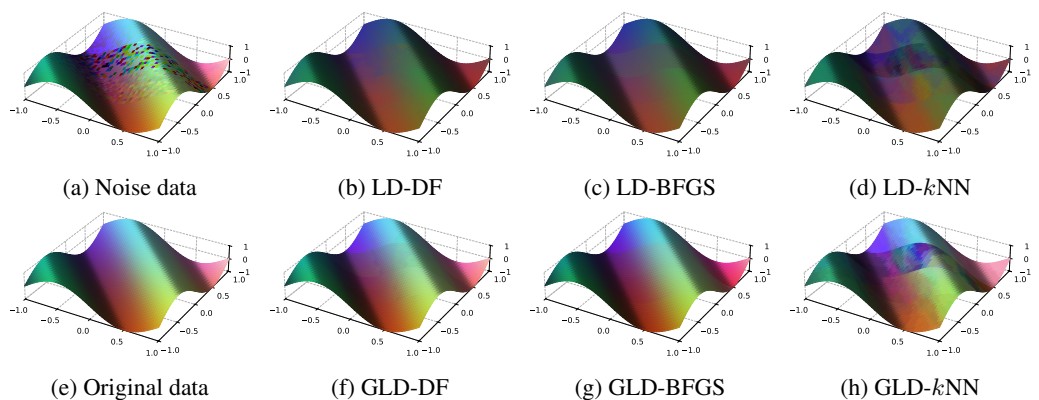

Figure 3: (Best viewed in color) Visualized results of the robustness testing.

2.8 and 2.9, respectively. GLD effectively overcomes these deficiencies, yielding substantial performance gains in these tasks, while at the same time maintaining high accuracy in LD prediction.

Secondly, regarding the adapted algorithms, although their relative performance on the LDL task can be roughly ordered as LD-Delta ≻ LD-DF ≻ LD-LRR, the ranking changes after our adaptation, becoming GLD-DF ≻ GLD-Delta ≻ GLD-LRR. A possible explanation is that we do not introduce any specific optimizations tailored for GLD prediction. As a result, the more general ensemble framework demonstrates stronger adaptability and delivers superior performance.

**The real-world datasets** In most cases, GLD methods still outperform their corresponding LD baseline methods. Regarding the methods in Section 3.2, although they achieve impressive results on the artificial dataset, their performance on real-world datasets is notably less satisfactory. For example, on the `artf` dataset, GLD-SVR achieves an average rank of 2.86 among 12 methods, whereas on `jura` and `wq` it only reaches 5.71 and 8.00, respectively. On the `wq` dataset, the performance of inter-sample ranking is poor, due to the discrete nature of the target data values, which leads to misjudgments in tie cases. The remaining experimental results are shown in Appendix C.

**Robustness analysis** We use the squared Mahalanobis distance to guide the learning of GLD, where the covariance matrix not only captures pairwise label correlations but also, to some extent, serves as a noise-robust regularization term. This is an aspect that needs to be verified. Therefore, we conduct robustness tests similar to (Li et al., 2025), exploiting the same artificial dataset mentioned above. The difference is that we use GLDs as the original data for GLD methods.[2] From bottom to top, from left to right, the four types of processing in Figure 3a are: (1) no treatment; (2) applying Gaussian noise (He et al., 2024); (3) randomly setting description degrees to zero (Xu & Zhou, 2017); (4) randomly emphasizing description degrees (Kou et al., 2023). These treatments correspond to different possible types of noise in label distributions. The prediction results of representative methods are shown in Figures 3b to 3d and 3f to 3h, and the ground truth, i.e., the original artificial data without noise, is shown in Figure 3e for comparison. The consistency of each prediction result reflects the robustness of each method. Figure 3 shows that methods like GLD-BFGS and GLD-DF predict accurately and the prediction results are not easily affected by noise.

## 5 LIMITATIONS & CONCLUSION

**Limitations** Despite these promising results, several limitations remain. In particular, the scenario when the semantics of the label space are continuous rather than discrete has not been explored. More applications of GLD are yet to be explored.

**Conclusion** In this paper, we systematically analyze the limitations of LD, and propose GLD as as a more *unified*, *versatile*, and *faithful* representation of label ambiguity. GLD can recover raw data while preserving inter-sample order consistency, derive other label forms without information loss, and naturally capture out-of-distribution samples and negative label correlations.

---

[2]Differences in color depth of the results also show limitations of the LD representation (Remark 2.9).

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

## A PROOFS AND DERIVATIONS

Here, we provide the detailed proofs and derivations omitted in the main paper.

### A.1 PROOF OF THEOREM 3.3

*Proof.* We first introduce the data processing inequality.

**Lemma A.1** (Beaudry & Renner (2012)). *Let $\mathfrak{A}$, $\mathfrak{B}$, and $\mathfrak{C}$ be random variables. If $\mathfrak{A} \to \mathfrak{B} \to \mathfrak{C}$ is a Markov chain, then we have*

$$I(\mathfrak{A}; \mathfrak{C}) \leq I(\mathfrak{A}; \mathfrak{B}). \tag{17}$$

*The equality holds if $\mathfrak{A} \to \mathfrak{C} \to \mathfrak{B}$ is also a Markov chain.*

According to Assumption 2.1 and Definition 3.2, we have

$$\mathfrak{G} \xrightarrow{\eta^{-1}} \mathfrak{P} \xrightarrow{\text{proj}} \mathfrak{D}. \tag{18}$$

According to Lemma A.1, we have

$$I(\mathfrak{G}; \mathfrak{D}) \leq I(\mathfrak{G}; \mathfrak{P}). \tag{19}$$

$\eta^{-1}(\cdot)$ is bijective. The mutual information is invariant under bijective transformations, i.e.,

$$I(\mathfrak{G}; \mathfrak{D}) = I(\mathfrak{P}; \mathfrak{D}). \tag{20}$$

The mutual information is symmetric, i.e.,

$$I(\mathfrak{G}; \mathfrak{P}) = I(\mathfrak{P}; \mathfrak{G}). \tag{21}$$

Combining Equations (19) to (21) gives the desired inequality.

If there exists an equality case, $\mathfrak{G} \to \mathfrak{D} \to \mathfrak{P}$ is also a Markov chain. Note that $\mathfrak{G} \to \mathfrak{D}$ has already been guaranteed by Equation (18). If $\mathfrak{D} \to \mathfrak{P}$ holds, then $\exists_\psi (\forall_{\boldsymbol{q} \in \mathcal{Q}} (\exists_{\boldsymbol{d} \in \Delta^{c-1}} (\boldsymbol{q} = \psi(\boldsymbol{d}))))$ is a tautology, where $\psi(\cdot)$ is invertible since $\boldsymbol{d}$ can be determined by $\boldsymbol{q}$. The above condition is equivalent to $\forall_{\boldsymbol{q} \in \mathcal{Q}} (\boldsymbol{q} \in \psi(\Delta^{c-1}))$, i.e., $\mathcal{Q} \subseteq \psi(\Delta^{c-1})$. □

### A.2 DERIVATION OF EQUATION (14)

Let $\boldsymbol{h}_i = \boldsymbol{W} \boldsymbol{x}_i$, $\tilde{\boldsymbol{g}}_i = \tanh(\boldsymbol{h}_i)$, and $\boldsymbol{e}_i = \boldsymbol{g}_i - \tilde{\boldsymbol{g}}_i$ for all $i \in [n]$. The quadratic term in the loss is $\boldsymbol{e}_i^\top \boldsymbol{\Sigma}^{-1} \boldsymbol{e}_i$, where $\boldsymbol{\Sigma}^{-1}$ is symmetric. Its gradient with respect to $\boldsymbol{e}_i$ is

$$\frac{\partial \boldsymbol{e}_i^\top \boldsymbol{\Sigma}^{-1} \boldsymbol{e}_i}{\partial \boldsymbol{e}_i} = (\boldsymbol{\Sigma}^{-1} + (\boldsymbol{\Sigma}^{-1})^\top) \boldsymbol{e}_i = 2 \boldsymbol{\Sigma}^{-1} \boldsymbol{e}_i. \tag{22}$$

Since $\tanh(\cdot)$ is is applied element-wise, by the chain rule, we have

$$\frac{\partial \boldsymbol{e}_i}{\partial \tilde{\boldsymbol{g}}_i} \frac{\partial \tilde{\boldsymbol{g}}_i}{\partial \boldsymbol{h}_i} = -\text{diag}(\text{sech}^2(\boldsymbol{h}_i)). \tag{23}$$

Furthermore, the derivative of the loss with respect to the weight matrix $\boldsymbol{W}$ can be expressed as

$$\frac{\partial \ell}{\partial \boldsymbol{W}} = \frac{1}{n} \sum_{i \in [n]} \left( \frac{\partial \ell}{\partial \boldsymbol{h}_i} \right) \boldsymbol{x}_i^\top. \tag{24}$$

Combining Equations (22) to (24), we arrive at the final expression:

$$\frac{\partial \ell}{\partial \boldsymbol{W}} = -\frac{2}{n} \sum_{i \in [n]} ((\boldsymbol{\Sigma}^{-1}(\boldsymbol{g}_i - \tanh(\boldsymbol{W} \boldsymbol{x}_i))) \odot \text{sech}^2(\boldsymbol{W} \boldsymbol{x}_i)) \boldsymbol{x}_i^\top. \tag{25}$$

### A.3 PROOF OF THEOREM 3.6

*Proof.* Under Assumption 2.2, the LD prediction of LD-$k$NN for an unknown sample $\tilde{x}$ is given by:

$$\tilde{d} = \frac{1}{k} \sum_{x_i \in \mathcal{N}(\tilde{x})} d_i = \frac{1}{k} \sum_{x_i \in \mathcal{N}(\tilde{x})} \frac{q_i}{\sum_{j \in [c]} (q_i)_j}. \tag{26}$$

The LD prediction of GLD-$k$NN for $\tilde{x}$ is given by:

$$\tilde{d}' = \frac{\sum_{x_i \in \mathcal{N}(\tilde{x})} q_i}{\sum_{j \in [c]} (\sum_{x_i \in \mathcal{N}(\tilde{x})} q_i)_j}. \tag{27}$$

Define $s_i = \sum_{j \in [c]} (q_i)_j > 0$ for all $i \in [n]$. Assume that each observed raw vector $q_i$ can be decomposed into a scaled ground-truth distribution $d^*$ perturbed by noise:

$$q_i = s_i d^* + \varepsilon_i, \tag{28}$$

where $\varepsilon_i$ denotes the noise term. Thus,

$$d_i = \frac{q_i}{s_i} = d^* + \frac{\varepsilon_i}{s_i}. \tag{29}$$

For LD-$k$NN, we have

$$\tilde{d} - d^* = \frac{1}{k} \sum_{x_i \in \mathcal{N}(\tilde{x})} \left( d^* + \frac{\varepsilon_i}{s_i} \right) - d^* = \frac{1}{k} \sum_{x_i \in \mathcal{N}(\tilde{x})} \frac{\varepsilon_i}{s_i}. \tag{30}$$

For GLD-$k$NN, we have

$$\tilde{d}' - d^* = \frac{\sum_{x_i \in \mathcal{N}(\tilde{x})} (s_i d^* + \varepsilon_i)}{\sum_{x_i \in \mathcal{N}(\tilde{x})} s_i} - d^* = \frac{\sum_{x_i \in \mathcal{N}(\tilde{x})} \varepsilon_i}{\sum_{x_i \in \mathcal{N}(\tilde{x})} s_i}. \tag{31}$$

Assume that the noise vectors $\varepsilon_i$ are independent with zero mean and isotropic covariance, i.e., $\mathbb{E}[\varepsilon_i] = \mathbf{0}$ and $\mathbb{V}[\varepsilon_i] = \sigma^2 I$. The variance of LD-$k$NN is

$$\mathbb{V}[d_j - d_j^*] = \frac{\sigma^2}{k^2} \sum_{x_i \in \mathcal{N}(\tilde{x})} \frac{1}{s_i^2}. \tag{32}$$

The variance of GLD-$k$NN is

$$\mathbb{V}[d_j' - d_j^*] = \frac{k\sigma^2}{\left( \sum_{x_i \in \mathcal{N}(\tilde{x})} s_i \right)^2}. \tag{33}$$

By applying the QM-HM inequality to the sequence $1/s$, we have

$$\frac{k^2}{\left( \sum_{x_i \in \mathcal{N}(\tilde{x})} s_i \right)^2} \leq \frac{\left( \sum_{x_i \in \mathcal{N}(\tilde{x})} 1/s_i \right)^2}{k}. \tag{34}$$

By applying the AM-QM inequality to the sequence $1/s$, we have

$$\frac{\left(\sum_{\boldsymbol{x}_i \in \mathcal{N}(\tilde{\boldsymbol{x}})} 1/s_i\right)^2}{k^2} \leq \frac{\sum_{\boldsymbol{x}_i \in \mathcal{N}(\tilde{\boldsymbol{x}})} 1/s_i^2}{k}. \tag{35}$$

Combining Equations (34) and (35) yields

$$\frac{k}{\left(\sum_{\boldsymbol{x}_i \in \mathcal{N}(\tilde{\boldsymbol{x}})} s_i\right)^2} \leq \frac{1}{k^2} \sum_{\boldsymbol{x}_i \in \mathcal{N}(\tilde{\boldsymbol{x}})} \frac{1}{s_i^2}. \tag{36}$$

Substituting Equations (32) and (33) into this relation directly leads to

$$\mathbb{V}[d_j' - d_j^*] \leq \mathbb{V}[d_j - d_j^*]. \tag{37}$$

The equality holds when $\forall_{\boldsymbol{x}_i, \boldsymbol{x}_j \in \mathcal{N}(\tilde{\boldsymbol{x}})}(s_i = s_j)$ is a tautology.

$\square$

### A.4 PROOF OF THEOREM 3.7

*Proof.* The LL prediction of LL-$k$NN for an unknown sample $\tilde{\boldsymbol{x}}$ is given by:

$$
\begin{aligned}
\tilde{\boldsymbol{l}} &= \left[\!\!\left[ \frac{1}{k} \left( \sum_{\boldsymbol{x}_i \in \mathcal{N}(\tilde{\boldsymbol{x}})} \boldsymbol{l}_i \right) \geq \frac{1}{2} \right]\!\!\right] \\
&= \left[\!\!\left[ \frac{1}{k} \left( \sum_{\boldsymbol{x}_i \in \mathcal{N}(\tilde{\boldsymbol{x}})} [\![\boldsymbol{q}_i \geq \boldsymbol{\tau}']\!] \right) \geq \frac{1}{2} \right]\!\!\right] \\
&= \left[\!\!\left[ \left( \sum_{\boldsymbol{x}_i \in \mathcal{N}(\tilde{\boldsymbol{x}})} 2\boldsymbol{\tau}' \odot [\![\boldsymbol{q}_i \geq \boldsymbol{\tau}']\!] \right) \geq k\boldsymbol{\tau}' \right]\!\!\right].
\end{aligned} \tag{38}
$$

The LL prediction of GLD-$k$NN for $\tilde{\boldsymbol{x}}$ is given by:

$$
\begin{aligned}
\tilde{\boldsymbol{l}}' &= \left[\!\!\left[ \frac{1}{k} \left( \sum_{\boldsymbol{x}_i \in \mathcal{N}(\tilde{\boldsymbol{x}})} \boldsymbol{g}_i \right) \geq \boldsymbol{0} \right]\!\!\right] \\
&= \left[\!\!\left[ \frac{1}{k} \left( \sum_{\boldsymbol{x}_i \in \mathcal{N}(\tilde{\boldsymbol{x}})} \boldsymbol{q}_i \right) \geq \boldsymbol{\tau}' \right]\!\!\right] \\
&= \left[\!\!\left[ \left( \sum_{\boldsymbol{x}_i \in \mathcal{N}(\tilde{\boldsymbol{x}})} \boldsymbol{q}_i \right) \geq k\boldsymbol{\tau}' \right]\!\!\right].
\end{aligned} \tag{39}
$$

The only distinction between the two methods lies in how the target vectors of the $k$ nearest neighbors are aggregated prior to thresholding. If $\mathbb{V}[l_j' - l_j^*] \leq \mathbb{V}[l_j - l_j^*]$, then $\mathbb{V}[q_j] \leq \mathbb{V}[2\tau_j'[\![q_j \geq \tau_j']\!]]$, i.e.,

$$
\begin{aligned}
\mathbb{V}[q_j] &\leq 4(\tau_j')^2 \mathbb{P}(q_j \geq \tau_j')(1 - \mathbb{P}(q_j \geq \tau_j')) \\
\sqrt{\mathbb{V}[q_j]} &\leq 2|\tau_j'|\sqrt{\mathbb{P}(q_j \geq \tau_j')(1 - \mathbb{P}(q_j \geq \tau_j'))}.
\end{aligned} \tag{40}
$$

$\square$

## A.5 PROOF OF THEOREM 3.8

*Proof.* Define $\phi(\boldsymbol{u},\, \boldsymbol{v})$ as $\mathrm{M}^2(\boldsymbol{u}, \tanh(\boldsymbol{v}))$. Let $\boldsymbol{y}_1 = \tanh(\boldsymbol{v}_1)$ and $\boldsymbol{y}_2 = \tanh(\boldsymbol{v}_2)$. Then, we have

$$
\begin{aligned}
|\phi(\boldsymbol{u},\, \boldsymbol{v}_1) - \phi(\boldsymbol{u},\, \boldsymbol{v}_2)| &= |\mathrm{M}(\boldsymbol{u},\, \boldsymbol{y}_1) - \mathrm{M}(\boldsymbol{u},\, \boldsymbol{y}_2)| \\
&= |(\boldsymbol{u} - \boldsymbol{y}_1)^\top \boldsymbol{\Sigma}^{-1}(\boldsymbol{u} - \boldsymbol{y}_1) - (\boldsymbol{u} - \boldsymbol{y}_2)^\top \boldsymbol{\Sigma}^{-1}(\boldsymbol{u} - \boldsymbol{y}_2)| \\
&= |((\boldsymbol{u} - \boldsymbol{y}_1) + (\boldsymbol{u} - \boldsymbol{y}_2))^\top \boldsymbol{\Sigma}^{-1}((\boldsymbol{u} - \boldsymbol{y}_1) - (\boldsymbol{u} - \boldsymbol{y}_2))| \\
&= |(2\boldsymbol{u} - \boldsymbol{y}_1 - \boldsymbol{y}_2)^\top \boldsymbol{\Sigma}^{-1}(\boldsymbol{y}_2 - \boldsymbol{y}_1)|.
\end{aligned}
\tag{41}
$$

According to Cauchy-Schwarz inequality,

$$
|(2\boldsymbol{u} - \boldsymbol{y}_1 - \boldsymbol{y}_2)^\top \boldsymbol{\Sigma}^{-1}(\boldsymbol{y}_2 - \boldsymbol{y}_1)| \leq \|2\boldsymbol{u} - \boldsymbol{y}_1 - \boldsymbol{y}_2\| \cdot \|\boldsymbol{\Sigma}^{-1}\| \cdot \|\boldsymbol{y}_2 - \boldsymbol{y}_1\|.
\tag{42}
$$

And we have

$$
\|2\boldsymbol{u} - \boldsymbol{y}_1 - \boldsymbol{y}_2\| \leq 2\|\boldsymbol{u}\| + \|\boldsymbol{y}_1\| + \|\boldsymbol{y}_2\| = 4\sqrt{c}.
\tag{43}
$$

Combine Equations (41) to (43), which shows that $\phi(\boldsymbol{u},\, \cdot)$ is $L$-Lipschitz, i.e., for $\boldsymbol{v}_1,\, \boldsymbol{v}_2 \in \mathbb{R}^c$,

$$
|\phi(\boldsymbol{u},\, \boldsymbol{v}_1) - \phi(\boldsymbol{u},\, \boldsymbol{v}_2)| \leq L\|\tanh(\boldsymbol{v}_1) - \tanh(\boldsymbol{v}_2)\|,
\tag{44}
$$

where $L = 4\sqrt{c}\|\boldsymbol{\Sigma}^{-1}\|$. The subsequent proof follows a process similar to (Wang & Geng, 2019b). Recall the definition of Rademacher complexity, i.e.,

$$
\hat{\mathfrak{R}}_n(\mathrm{M}^2 \circ \tanh \circ \mathcal{H} \circ \mathcal{S}) = \mathbb{E}\left[\sup_{h \in \mathcal{H}} \frac{1}{n} \sum_{i \in [n]} \phi(\boldsymbol{g}_i,\, h(\boldsymbol{x}_i))\nu_i\right],
\tag{45}
$$

where $\mu_i$ are $n$ i.i.d. rademacher random variables with $\mathbb{P}(\nu_i = 1) = \mathbb{P}(\nu_i = -1) = 1/2$ for all $i \in [n]$. And according to (Maurer, 2016), with $\phi(\boldsymbol{u},\, \cdot)$ being $4\sqrt{c}\|\boldsymbol{\Sigma}^{-1}\|$-Lipschitz, we have

$$
\mathbb{E}\left[\sup_{h \in \mathcal{H}} \frac{1}{n} \sum_{i \in [n]} \phi(\boldsymbol{g}_i,\, h(\boldsymbol{x}_i))\nu_i\right] \leq 4\sqrt{2c}\|\boldsymbol{\Sigma}^{-1}\|\mathbb{E}\left[\sup_{h \in \mathcal{H}} \frac{1}{n} \sum_{i \in [n]} \sum_{j \in [c]} \nu_{i,\, j} h_j(\boldsymbol{x}_i)\right],
\tag{46}
$$

where $h_j(\cdot)$ is the $j$-th component of $h(\cdot)$ and $\nu_{i,\, j}$ are $n \times c$ i.i.d. rademacher random variables. Suppose all $\mathcal{H}_j$s be classes of functions for network-based architecture, then $\mathcal{H} = \oplus_{j \in [c]}\mathcal{H}_j = \{\boldsymbol{x} \mapsto h(\boldsymbol{x})\}$, and we have

$$
\begin{aligned}
\mathbb{E}\left[\sup_{h \in \oplus_{j \in [c]}\mathcal{H}_j} \frac{1}{n} \sum_{i \in [n]} \sum_{j \in [c]} \nu_{i,\, j} h_j(\boldsymbol{x}_i)\right] &\leq \sum_{j \in [c]} \mathbb{E}\left[\sup_{h_j \in \mathcal{H}_j} \frac{1}{n} \sum_{i \in [n]} \nu_{i,\, j} h_j(\boldsymbol{x}_i)\right] \\
&\leq \sum_{j \in [c]} \hat{\mathfrak{R}}_n(\mathcal{H}_j \circ \mathcal{S}).
\end{aligned}
\tag{47}
$$

Combine Equations (45) to (47), which finishes proof of Theorem 3.8.

$\square$

## A.6 Proof of Theorem 3.9

*Proof.* We first introduce the following two lemmas.

**Lemma A.2** (Bartlett & Mendelson (2003); Mohri et al. (2012)). *Let $\mathcal{H}$ be a family of functions. For a loss function $\ell$ bounded by $\mu$, then for any $\delta > 0$, with probability at least $1 - \delta$, for all $h \in \mathcal{H}$ such that*

$$\vartheta_{\mathcal{D}}(h) \leq \hat{\vartheta}_{\mathcal{S}}(h) + 2\hat{\mathfrak{R}}(\ell \circ \mathcal{H} \circ \mathcal{S}) + 3\mu\sqrt{\frac{\log(2/\delta)}{2n}}. \tag{48}$$

**Lemma A.3** (Bartlett & Mendelson (2003); Gao & Zhou (2016)). *Define class of functions $\mathcal{H}_j = \{\boldsymbol{x} \mapsto \boldsymbol{w}_j \cdot \boldsymbol{x}\}$, where $\|\boldsymbol{w}_j\|_2 \leq \xi$ for all $j \in [c]$. We have*

$$\hat{\mathfrak{R}}_n(\mathcal{H}_j \circ \mathcal{S}) \leq \frac{\xi \max_{i \in [n]} \|\boldsymbol{x}_i\|}{\sqrt{n}}. \tag{49}$$

Substituting Theorem 3.8 and Lemma A.3 into Lemma A.2, and Theorem 3.9 follows directly.

$\square$

## B More insights about Table 1

A more detailed version of Table 1 is presented in Table 3.

Table 3: Conversion capabilities among label ambiguity representations.

| From | To | | | | | |
|------|-----|-----|-----|-----|-----|-----|
| | GLD | LD | TL | LL | LR | SL |
| GLD | $\dagger_1$ | Eq. (50) | Eq. (51) | Eq. (52) | Eq. (53) | Eq. (54) |
| LD | | $\dagger_2$ | $\dagger_3$ | $\dagger_4$ | Eq. (55) | Eq. (56) |
| TL | | $\dagger_5$ | | $\dagger_6$ | $\dagger_7$ | $\dagger_8$ |
| LL | | $\dagger_9$ | | $\dagger_{10}$ | $\dagger_{11}$ | $\dagger_{12}$ |
| LR | | $\dagger_{13}$ | | | | Eq. (57) |
| SL | | | | $\dagger_{14}$ | | $\dagger_{15}$ |

GLD can be transformed into other label ambiguity representations through Equations (50) to (54). The conversion from GLD to LD relies on the capability of GLD to recover the raw data:

$$\text{GLD2LD}(\boldsymbol{g}) = \text{proj}\left(\frac{(\boldsymbol{g}+1) \odot (\boldsymbol{b}-\boldsymbol{a})}{2} + \boldsymbol{a}\right). \tag{50}$$

**Definition B.1** (Ternary label). The TL is a ternary vector $\boldsymbol{t} \in \{-1, 0, 1\}^c$, where $t_j = 1$ indicates that the $j$-th label $y_j \in \mathcal{Y}$ is positively associated with the sample, $t_j = -1$ indicates a negative association, and $t_j = 0$ means the label is irrelevant.

Based on Definition B.1 and Definition 2.3, GLD can be naturally converted into TL and LL by

$$\text{GLD2TL}(\boldsymbol{g}; \boldsymbol{\tau}^{(+)}, \boldsymbol{\tau}^{(-)})_j = \begin{cases} [\![g_j > \tau_j^{(+)}]\!], & g_j \geq \tau_j^{(-)}, \\ -1, & g_j < \tau_j^{(-)}, \end{cases} \tag{51}$$

$$\text{GLD2LL}(\boldsymbol{g}; \boldsymbol{\tau})_j = [\![g_j \geq \tau_j]\!], \tag{52}$$

respectively. To ensure consistency with the raw data, the threshold vectors are generally set as $\boldsymbol{\tau}^{(+)} = 1/3 \mathbf{1}$, $\boldsymbol{\tau}^{(-)} = -1/3 \mathbf{1}$, and $\boldsymbol{\tau} = \mathbf{0}$. Next, we define LR in a simplified form.

**Definition B.2** (Label ranking). The LR is defined as a label vector $\boldsymbol{r} \in \mathcal{Y}^c$, where each element corresponds to a label $y \in \mathcal{Y}$ and the order reflects their relative preference or relevance to the sample, and $r_j \neq r_k$ for all $j \neq k$.

Both GLD and LD can then be converted into LR and SL via $\arg\text{sort}$ and $\arg\max$, since the necessary information is preserved in their respective data structures:

$$\text{GLD2LR}(\boldsymbol{g}) = \underset{y \in \mathcal{Y}}{\arg\text{sort}}(\boldsymbol{g}), \tag{53}$$

$$\text{GLD2SL}(\boldsymbol{g}) = \underset{y \in \mathcal{Y}}{\arg\max}(\boldsymbol{g}), \tag{54}$$

$$\text{LD2LR}(\boldsymbol{d}) = \underset{y \in \mathcal{Y}}{\arg\text{sort}}(\boldsymbol{d}), \tag{55}$$

$$\text{LD2SL}(\boldsymbol{d}) = \underset{y \in \mathcal{Y}}{\arg\max}(\boldsymbol{d}). \tag{56}$$

Moreover, LR can be further converted into SL by selecting the top-ranked label:

$$\text{LR2SL}(\boldsymbol{r}) = r_1. \tag{57}$$

The remaining white area in Table 3 corresponds to lossy/impossible conversions. For example, the conversion from LD to LL, i.e., $\dagger_4$, has been proven to be inconsistent with the raw data, as detailed in Section 2. Similarly, $\dagger_3$ also represents a lossy conversion. In the case of $\dagger_6$, irrelevant labels cannot be reassigned in a reasonable manner. Likewise, conversions such as $\dagger_7$, $\dagger_8$, $\dagger_{11}$ and $\dagger_{12}$ are not feasible, since labels with the same discrete value do not possess an inherent order.

Next, we provide further insights into Table 3. The gray area reveals potential learning paradigms. The diagonal entries correspond to standard learning paradigms. For example, $\dagger_1$ corresponds precisely to the learning paradigm proposed in this paper, $\dagger_2$ refers to LDL (Geng, 2016) or incomplete/inaccurate LDL (Xu & Zhou, 2017; Kou et al., 2023), $\dagger_{10}$ refers to MLL (Zhang & Zhou, 2013) or partial multi-label learning (Xie & Huang, 2018), and $\dagger_{15}$ is for single-label classification and its variants. The lower triangular section comprises recovery/enhancement paradigms. For example, $\dagger_9$ represents label enhancement (Xu et al., 2019), $\dagger_{14}$ can be regarded as single positive label learning (Cole et al., 2021), and $\dagger_5$ and $\dagger_{13}$ correspond to (Lu & Jia, 2024) and (Lu & Jia, 2022), respectively.

Table 3 not only demonstrates the advantage of GLD in being fully compatible with existing label ambiguity representations, but also illustrates how our work links prior related studies.

## C  MORE DETAILS OF EXPERIMENTS

**The artificial dataset** In this dataset, $\boldsymbol{x}$ is of three-dimensional and there are three labels, i.e., $m = 3$ and $c = 3$. The corresponding GLD $\boldsymbol{g}$ of $\boldsymbol{x}$ is generated in the following ways:

$$t_k = ax_k + bx_k^2 + cx_k^3 + d \ (k = 1,\ 2,\ 3), \tag{58}$$

$$\varpi_1 = \tanh(\boldsymbol{w}_1^\top \boldsymbol{t}), \tag{59}$$

$$\varpi_2 = (1 - \lambda_1)\tanh(\boldsymbol{w}_2^\top \boldsymbol{t}) + \lambda_1 \varpi_1, \tag{60}$$

$$\varpi_3 = (1 - \lambda_2)\tanh(\boldsymbol{w}_3^\top \boldsymbol{t}) + \lambda_2 \varpi_2, \tag{61}$$

$$g_j = \frac{\varpi_j}{\varpi_1 + \varpi_2 + \varpi_3} \ (j = 1,\ 2,\ 3). \tag{62}$$

Note that Equations (60) and (61) deliberately make the relative degree of one label depend on those of other labels. The parameters in Equations (58) to (61) are set as $a = 2$, $b = 4$, $c = 1$, $d = 1$,

$\boldsymbol{w}_1 = \langle 0.4,\, 0.2,\, -1.0 \rangle^\top$, $\boldsymbol{w}_2 = \langle 0.2,\, 0.1,\, 0.4 \rangle^\top$, $\boldsymbol{w}_3 = \langle -0.1,\, 0.4,\, 0.2 \rangle^\top$, and $\lambda_1 = \lambda_2 = 0.01$. To generate the dataset, each component of $\boldsymbol{x}$ is uniformly sampled within the range $[-1,\ 1]_\mathbb{R}$, and then the GLD $\boldsymbol{g}$ corresponding to each $\boldsymbol{x}$ is calculated via Equations (58) to (62). In total, there are 200 data generated in this way.

**The real-world datasets** According to Definition 3.2 and Equation (50), we preprocess the raw data in $\prod_{j\in[c]}[a_j,\, b_j]_\mathbb{R}$ into GLDs in $[-1,\ 1]_\mathbb{R}^c$ and LDs in $\Delta^{c-1}$, thereby constructing real-world datasets for experiments, where $\boldsymbol{a}$ and $\boldsymbol{b}$ serve only as meta-data. Additional experimental results are provided in Table 4. It is important to note that for the jaf dataset, the raw data is shifted from the original $[1,5]\mathbb{R}$ interval to $[0,4]\mathbb{R}$ to ensure greater reasonability. Consequently, the reported metric values are not directly comparable to those presented in previous literature.

**Evaluation metrics** The Clark distance is one of the earliest metrics used for evaluating the task of LDL (Geng, 2016). It is sensitive to values approaching zero and, when the ground-truth LD is sparse, can reflect the sparsity of the predicted LD to some extent. Its definition is as follows:

$$\texttt{Clark}(\boldsymbol{u},\, \boldsymbol{v}) \triangleq \sqrt{\sum_{j\in[c]} \frac{(u_j - v_j)^2}{(u_j + v_j)^2}}. \tag{63}$$

With a slight abuse of notation, we use $\texttt{Clark}(\boldsymbol{D},\, \tilde{\boldsymbol{D}}) \triangleq \nicefrac{1}{n} \sum_{i\in[n]} \texttt{Clark}(\boldsymbol{d}_i,\, \tilde{\boldsymbol{d}}_i)$ to evaluate the performance on the test set, where $\boldsymbol{D}$ and $\tilde{\boldsymbol{D}}$ are the matrices of the ground-truth and predicted LDs for all $n$ test samples, respectively. The $\mu$ metric is proposed in (Li et al., 2025). It can be combined with existing metrics to quantify the extent to which samples are predicted as approximately correct. Its definition, when combined with the KLD, is as follows:

$$\mu_{\texttt{KLD}}(\boldsymbol{D},\, \tilde{\boldsymbol{D}}) \triangleq \frac{1}{\delta_0} \int_0^{\delta_0} \frac{1}{n} \sum_{i\in[n]} [\![\texttt{KLD}(\boldsymbol{d}_i \,||\, \tilde{\boldsymbol{d}}_i) \leq \delta]\!] \mathrm{d}\delta, \tag{64}$$

where

$$\texttt{KLD}(\boldsymbol{u} \,||\, \boldsymbol{v}) \triangleq \sum_{j\in[c]} u_j \log \frac{u_j}{v_j}, \tag{65}$$

and

$$\delta_0 = \frac{1}{n} \sum_{i\in[n]} \sum_{j\in[c]} (\boldsymbol{d}_i)_j \log(c(\boldsymbol{d}_i)_j) \tag{66}$$

is the worst-case KLD, calculated between the ground-truth LDs and the uniform vector.

For the MLL task, Hamming distance and subset accuracy are commonly used evaluation metrics (Zhang & Zhou, 2013). The Hamming distance is defined as

$$\texttt{Ham.}(\boldsymbol{u},\, \boldsymbol{v}) \triangleq \frac{1}{c} \sum_{j\in[c]} [\![u_j \neq v_j]\!]. \tag{67}$$

We further evaluate on the test set using $\texttt{Ham.}(\boldsymbol{L},\, \tilde{\boldsymbol{L}}) \triangleq \nicefrac{1}{n} \sum_{i\in[n]} \texttt{Ham.}(\boldsymbol{l}_i,\, \tilde{\boldsymbol{l}}_i)$, where $\boldsymbol{L}$ and $\tilde{\boldsymbol{L}}$ are the matrices of the ground-truth and predicted LLs for all $n$ test samples, respectively. The definition of subset error rate is given in Equation (2), and thus the subset accuracy on the test set is calculated as $\texttt{S.Acc.}(\boldsymbol{L},\, \tilde{\boldsymbol{L}}) \triangleq \nicefrac{1}{n} \sum_{i\in[n]} 1 - \texttt{S.Err.}(\boldsymbol{l}_i,\, \tilde{\boldsymbol{l}}_i)$.

To evaluate order consistency, we consider both intra-sample and inter-sample ranking metrics. For intra-sample ranking, we adopt Spearman's rank correlation coefficient (Jia et al., 2023), defined as

$$\texttt{Spear.}(\boldsymbol{u},\,\boldsymbol{v}) \triangleq 1 - \frac{6\sum_{j\in[c]}(\rho(u_j) - \rho(v_j))^2}{c(c-1)}, \tag{68}$$

where $\rho(\cdot)$ denotes the ranking function. The overall performance on the test set is then measured by $\texttt{Spear.}(\boldsymbol{U},\,\boldsymbol{V}) \triangleq 1/n \sum_{i\in[n]} \texttt{Spear.}(\boldsymbol{u}_i,\,\boldsymbol{v}_i)$.

For inter-sample ranking, we employ an averaged Kendall's tau statistic (type-A) across labels, given by $\texttt{Ken.}'(\boldsymbol{U},\,\boldsymbol{V}) \triangleq 1/c \sum_{j\in[c]} \texttt{Ken.}(\boldsymbol{u}_{\bullet j},\,\boldsymbol{v}_{\bullet j})$ where $\texttt{Ken.}(\cdot,\,\cdot)$ is defined in Equation (7).

For the task of OOD classification, we propose a newly designed evaluation metric, defined as

$$\texttt{OOD\,Err.}(\boldsymbol{g},\,\boldsymbol{v}) \triangleq \begin{cases} \begin{cases} [\![\forall_{j\in[c]}(v_j \leq 0)]\!], & \text{if } \boldsymbol{v} \text{ is GLD,} \\ [\![\sum_{j\in[c]} v_j \log(cv_j) \leq \delta_0/2]\!], & \text{if } \boldsymbol{v} \text{ is LD,} \\ [\![\arg\max_j(\boldsymbol{g}) = \arg\max_j(\boldsymbol{v})]\!], & \text{o/w,} \end{cases} & \forall_{j\in[c]}(g_j \leq 0), \\ & \text{o/w,} \end{cases} \tag{69}$$

where $\delta_0$ is defined in Equation (66), and $\boldsymbol{g}$ is the ground-truth GLD. We further evaluate on the test set using $\texttt{OOD\,Err.}(\boldsymbol{G},\,\boldsymbol{V}) \triangleq 1/n \sum_{i\in[n]} \texttt{OOD\,Err.}(\boldsymbol{g}_i,\,\boldsymbol{v}_i)$.

Table 4: Experimental results on datasets `scm1d`, `jaf`, `enb`, and `atp` formatted as (mean $\pm$ std).

| Algorithms | Clark ↓ | $\mu_{KLD}$ ↑ | Ham. ↓ | S. Acc. ↑ | Spear. ↑ | Ken.′ ↑ | OOD Err. ↓ | Avg. Rank |
|---|---|---|---|---|---|---|---|---|
| | | | | scm1d | | | | |
| LD-SVR | ●.0955±.002 | ●90.50%±.004 | ●.4544±.004 | ●00.09%±.001 | .9247±.003 | .3421±.007 | ●49.31%±.016 | 7.57 |
| GLD-SVR | .0845±.002 | 92.06%±.004 | .0731±.003 | 45.09%±.015 | .9369±.003 | .7150±.006 | 37.23%±.014 | 2.43 |
| LD-kNN | .0654±.002 | 94.05%±.003 | ●.4484±.004 | ●00.12%±.001 | .9540±.002 | .3617±.007 | ●38.23%±.014 | 4.29 |
| GLD-kNN | **.0653**±.002 | **94.07%**±.003 | **.0488**±.002 | **56.45%**±.016 | **.9541**±.002 | **.7843**±.006 | **32.13%**±.014 | **1.00** |
| LD-BFGS | ●.1059±.002 | ●88.05%±.005 | ●.4604±.004 | ●00.09%±.001 | ●.9104±.004 | ●.3165±.007 | ●48.83%±.015 | 10.1 |
| GLD-BFGS | .0991±.002 | 89.34%±.004 | .1088±.003 | 36.49%±.012 | .9182±.003 | .6627±.006 | 43.51%±.016 | 6.86 |
| LD-DF | ●.0959±.002 | ●89.87%±.004 | ●.4586±.004 | ●00.11%±.001 | .9260±.003 | ●.3290±.007 | ●45.62%±.014 | 7.43 |
| GLD-DF | .0923±.002 | 90.56%±.004 | .1002±.003 | 38.52%±.012 | .9265±.003 | .6880±.006 | 41.34%±.015 | 4.00 |
| LD-LRR | ●.1148±.002 | ●86.32%±.005 | ●.4615±.005 | ●00.07%±.001 | ●.8985±.004 | ●.3036±.007 | ●50.94%±.017 | 12.0 |
| GLD-LRR | .1083±.002 | 87.72%±.005 | .1339±.004 | 29.88%±.014 | .9058±.004 | .6327±.007 | 49.79%±.019 | 8.86 |
| LD-Delta | .0965±.002 | ●89.71%±.004 | ●.4602±.004 | ●00.09%±.001 | .9209±.003 | ●.3347±.007 | ●46.72%±.016 | 8.57 |
| GLD-Delta | .0960±.002 | 89.91%±.004 | .0815±.003 | 43.25%±.015 | .9222±.003 | .6737±.006 | 41.09%±.015 | 4.86 |
| | | | | jaf | | | | |
| LD-SVR | ●.6414±.061 | 50.14%±.060 | ●.3246±.040 | ●09.99%±.062 | .4821±.109 | ●.3538±.073 | ●51.41%±.103 | 9.71 |
| GLD-SVR | .6189±.056 | 51.40%±.069 | .2177±.037 | 23.74%±.084 | .4936±.107 | .3915±.077 | 44.37%±.101 | 5.43 |
| LD-kNN | .6185±.053 | 50.98%±.070 | ●.3142±.043 | ●08.54%±.058 | .5064±.093 | .3919±.078 | ●50.94%±.101 | 7.86 |
| GLD-kNN | .6145±.058 | 52.37%±.066 | .2146±.038 | 25.23%±.083 | .5280±.091 | .3883±.075 | 49.38%±.108 | 4.71 |
| LD-BFGS | ○.6297±.047 | ○51.43%±.089 | ●.2856±.047 | ●18.31%±.094 | .5084±.100 | .3984±.079 | 46.19%±.110 | 5.86 |
| GLD-BFGS | .6781±.054 | 47.83%±.091 | .2134±.039 | 26.74%±.092 | .5116±.096 | .4094±.069 | 45.51%±.116 | 5.00 |
| LD-DF | ○**.5740**±.049 | **57.70%**±.075 | ●.2913±.039 | ●15.88%±.074 | .5365±.084 | .4287±.075 | ●**40.79%**±.116 | 3.43 |
| GLD-DF | .5951±.054 | 56.25%±.076 | **.1967**±.037 | 32.26%±.104 | .5427±.095 | .4337±.064 | 44.56%±.101 | **1.71** |
| LD-LRR | .6993±.051 | 42.23%±.089 | ●.3137±.048 | ●16.10%±.083 | .4137±.103 | .3195±.085 | 51.86%±.109 | 10.7 |
| GLD-LRR | .7596±.212 | 39.83%±.135 | .2488±.085 | 25.31%±.118 | .3969±.149 | .3255±.125 | 51.28%±.138 | 9.57 |
| LD-Delta | ○.6152±.052 | ○52.75%±.090 | ●.3114±.048 | ●14.05%±.084 | .4704±.110 | .3734±.101 | ○44.26%±.114 | 6.43 |
| GLD-Delta | .6864±.084 | 45.68%±.092 | .2243±.039 | 28.71%±.086 | .4482±.102 | .3569±.078 | 47.42%±.089 | 7.57 |
| | | | | atp1d | | | | |
| LD-SVR | .1051±.011 | 77.83%±.035 | .4793±.016 | ●01.28%±.019 | .8200±.039 | ●.1885±.054 | ●52.17%±.069 | 6.57 |
| GLD-SVR | **.1021**±.012 | **78.12%**±.035 | **.0321**±.019 | **89.97%**±.051 | .8102±.038 | **.5191**±.050 | **09.00%**±.042 | **1.43** |
| LD-kNN | .1072±.014 | 76.44%±.035 | .4818±.015 | ●00.86%±.015 | .8262±.039 | ●.1755±.054 | ●51.84%±.069 | 7.71 |
| GLD-kNN | .1072±.013 | 76.52%±.035 | .0382±.021 | 89.01%±.051 | .8232±.038 | .4782±.058 | 11.26%±.048 | 3.43 |
| LD-BFGS | ●.1462±.016 | ●64.06%±.051 | .4815±.018 | ●01.91%±.024 | ●.7008±.061 | ●.1542±.050 | ●49.80%±.076 | 10.1 |
| GLD-BFGS | .1392±.013 | 65.81%±.048 | .0467±.026 | 86.34%±.057 | .7280±.051 | .4079±.053 | 11.28%±.051 | 7.14 |
| LD-DF | ●.1224±.013 | ●71.26%±.042 | .4801±.016 | ●01.82%±.022 | ●.7549±.053 | ●.1718±.049 | ●51.68%±.081 | 9.43 |
| GLD-DF | .1119±.012 | 75.01%±.040 | .0421±.023 | 87.65%±.055 | .7926±.041 | .4932±.053 | 10.93%±.048 | 4.57 |
| LD-LRR | ○.1163±.012 | ○74.04%±.038 | ●.4784±.016 | ●01.99%±.022 | ○.7586±.050 | .1763±.046 | ●47.57%±.072 | 7.71 |
| GLD-LRR | .2967±.055 | 17.03%±.123 | .1200±.046 | 81.70%±.080 | .4057±.163 | .1043±.079 | 15.78%±.087 | 9.43 |
| LD-Delta | ○.1053±.011 | ○77.64%±.036 | ●.4771±.017 | ●01.81%±.025 | .7926±.046 | ●.1903±.051 | ●51.48%±.074 | 6.14 |
| GLD-Delta | .1144±.018 | 74.45%±.064 | .0353±.024 | 89.01%±.050 | .7812±.052 | .4958±.055 | 09.36%±.046 | 4.29 |
| | | | | enb | | | | |
| LD-SVR | ○.0209±.001 | ○84.06%±.032 | ●.5306±.013 | ●00.26%±.005 | .7157±.076 | ●.0309±.010 | ●58.31%±.049 | 8.14 |
| GLD-SVR | .0242±.003 | 80.31%±.030 | .1235±.028 | 80.24%±.047 | .6891±.078 | .4556±.054 | 14.19%±.039 | 8.00 |
| LD-kNN | ○**.0148**±.001 | ○90.22%±.022 | ●.5172±.013 | ●01.60%±.013 | .7760±.069 | ●.0490±.011 | ●57.68%±.053 | 4.71 |
| GLD-kNN | .0160±.002 | 89.10%±.019 | **.0564**±.021 | 89.57%±.038 | .8226±.058 | **.7060**±.039 | 12.61%±.038 | **1.86** |
| LD-BFGS | ○.0213±.002 | ○82.48%±.030 | ●.5293±.014 | ●00.39%±.007 | .7111±.083 | ●.0315±.010 | ●61.37%±.047 | 8.71 |
| GLD-BFGS | .0238±.002 | 79.13%±.035 | .0724±.024 | 88.26%±.038 | .7309±.072 | .5535±.049 | **11.30%**±.039 | 5.14 |
| LD-DF | ○.0203±.002 | ○83.31%±.032 | ●.5172±.013 | ●01.60%±.013 | .7624±.077 | ●.0359±.011 | ●58.66%±.053 | 6.00 |
| GLD-DF | .0217±.002 | 81.80%±.030 | .0691±.020 | 86.59%±.039 | .7010±.078 | .5669±.058 | 11.73%±.039 | 5.86 |
| LD-LRR | ○.0213±.002 | ○82.49%±.030 | ●.5293±.014 | ●00.39%±.007 | .7111±.083 | ●.0315±.010 | ●61.37%±.047 | 8.57 |
| GLD-LRR | .0272±.024 | 77.68%±.089 | .0729±.027 | 88.18%±.047 | .7187±.084 | .5553±.049 | 11.34%±.039 | 6.14 |
| LD-Delta | ○.0211±.002 | ○82.30%±.032 | ●.5297±.014 | ●00.35%±.007 | .7146±.087 | ●.0285±.011 | ●62.89%±.054 | 9.43 |
| GLD-Delta | .0232±.002 | 80.12%±.039 | .0744±.025 | 88.18%±.039 | .7222±.079 | .5688±.048 | 11.58%±.039 | 5.43 |

