# OpenReview forum: "Learning Generalized Label Distributions"
_ICLR.cc/2026/Conference — Submitted to ICLR 2026_

### Official Review · Reviewer_Jzzj · 2025-10-28

**Soundness:** 1
**Presentation:** 3
**Contribution:** 1
**Rating:** 2
**Confidence:** 4

**Summary:**

The manuscript proposed generalized label distribution to solve the issues faced by current label distribution methods, such as inconsistency with raw data, distortion of inter-sample order, and limited applicability. And they demonstrate the effectiveness through
both theoretical analysis and extensive experiments.

**Strengths:**

The math seems solid. The writing is clear and easy to follow. But the significance is very limited.

**Weaknesses:**

The motivation is not sound or meaningless. What is the point to propose a generalized label distribution? At the beginning, the authors claim the label ambiguity exists, which means the label ambiguity is a problem while label distribution is a kind of method to solve the label ambiguity. But the authors take the label distribution as a research problem which is misleading. Just as written in the limitations by the authors, "applications of GLD are yet to be explored". It is meaningless to propose a generalized label distribution as a solution to solve the drawbacks of label distribution while no application setting need it; thus, the contribution is very limited.

**Questions:**

What is the point to propose a generalized label distribution?

---

> ### Author Response · Authors · 2025-11-16
> **Response to the comments of Reviewer Jzzj (Part I)**
>
> We appreciate the Reviewer's comments. We have carefully considered the comments and have provided detailed clarifications and additional references below to better support our motivation and research premises, which are well-established within our field.
>
> ---
>
> **Comment 1:** "No application setting needs LD/GLD."
>
> **Response:** We agree with the Reviewer that exploring downstream applications is an important future direction, and we have stated so in our manuscript. However, we respectfully disagree that there is no application setting needs LD/GLD. In fact, the limitations of LD in *existing real-world scenarios* constitute the primary motivation for developing GLD. More importantly, whenever the label space exhibits discrete semantics, GLD can naturally adapt to all existing LD applications. Below, we illustrate this with concrete examples:
>
> + **Facial emotion recognition:** Human emotions, particularly some negative emotions, are inherently complex and often composed of multiple primary emotional components. Consequently, many studies represent them as emotion distributions [1-3]. However, current LD carries a strong assumption that all labels are relevant to the sample and their description degrees are mutually inhibitory, which can distort the true information (the second issue in **Remark 2.9**). Our GLD framework relaxes these constraints, thereby enabling a more faithful and nuanced representation. This is empirically validated on the **jaf** dataset for facial emotion recognition in our experiments.
> + **Compositional data prediction:** LD has been used in various scientific applications. NASA, for example, has used LD to determine the chemical compositions of Martian meteorites [4]. Ref [5] disassembles video actions into constituent units and leverages LD to handle inter-unit ambiguity for effective video parsing. Ref [6] employs LD on crowd-sourced data to predict a selection distribution, capturing both audience consensus and input ambiguity. However, LD is inherently unable to recover the absolute magnitudes underlying compositional data (the first issue in **Remark 2.9**). The proposed GLD mapping is bijective so that the raw data can be accurately reconstructed. In our experiments, the **jura**, **wq**, and **enb** datasets all correspond to this compositional data prediction application.
>
> In recent years, LD has found applications in various medical tasks, including lesion grading [7], breast tumor cellularity estimation [8], bone age assessment [9], and blood pressure prediction [10].
>
> When we state that the applications of GLD are yet to be explored, we specifically refer to *extending beyond the traditional scope of LD*. For instance, modeling temporal dynamics such as upward and downward trends in time-series data, which are not feasible under traditional LD. In summary, while LD has established a solid foundation, GLD generalizes its capabilities, enabling a broader range of applications and more flexible modeling of label semantics.

---

> ### Author Response · Authors · 2025-11-16
> **Response to the comments of Reviewer Jzzj (Part II)**
>
> (Cont'd)
>
> **Comment 2:** "Taking LD as a research problem is misleading."
>
> **Response:** We respectfully disagree with the Reviewer's concern. Learning LD is both necessary and meaningful, as it provides a nuanced framework to model *label ambiguity/polysemy* and captures richer information than traditional single-label or multi-label formulations. Label distribution learning (LDL) has been an established learning paradigm for over a decade [11] and has been widely recognized in the community, with substantial literature exploring its interpretability [12], theoretical properties [13], and application potential [1-10]. Moreover, LDL has given rise to a variety of research directions that have brought substantial value to the community, including but not limited to:
>
> + **Label enhancement** [14]: Studying how to recover LDs from logical labels in order to improve the performance of multi-label learning.
> + **Incomplete LDL** [15]: Modeling under incomplete information and exploring label correlations as well as the low-rank properties of LDs or logical labels.
> + **LDL for classification** [16]: Leveraging margin theory to investigate how label distributions can enhance classifier performance.
>
> These directions demonstrate that LDL is not merely an abstract problem, but a research paradigm with both theoretical depth and practical significance.
>
> **Comment 3:** What is the point to propose the GLD?
>
> **Response:** The motivation for proposing GLD becomes clear once the broad applicability and the research significance of LD are established:
>
> *Our proposed GLD is designed as a unified framework for modeling label polysemy, addressing the inherent limitations of conventional LD, further extending its representational flexibility and application scope.*
>
> This correction is particularly urgent because existing studies tend to overestimate the expressive power of conventional LD, for example, assuming that LD can be directly converted into logical labels and exploited [17, 18], which we demonstrate is not theoretically justified. By addressing these limitations, GLD not only provides a more accurate and flexible representation of label semantics, but also paves the way for subsequent research directions to proceed more effectively and correctly.
>
> ---
>
> **Conclusion:** We sincerely invite the Reviewer to re-evaluate our manuscript. Please let us know if our responses have addressed your concerns, and we welcome any further questions or comments at any time.
>
> [1] Jia et al. Facial emotion distribution learning by exploiting low-rank label correlations locally. *CVPR*, 2019.
>
> [2] Chen et al. Label distribution learning on auxiliary label space graphs for facial expression recognition. *CVPR*, 2020.
>
> [3] Shao et al. Self-paced label distribution learning for in-the-wild facial expression recognition. *ACM MM*, 2022.
>
> [4] Morrison et al. Predicting multi-component mineral compositions in Gale crater, Mars with label distribution learning. *American Geophysical Union, Fall Meeting*, 2018.
>
> [5] Geng and Ling. Soft video parsing by label distribution learning. *AAAI*, 2017.
>
> [6] Shirani et al. Learning emphasis selection for written text in visual media from crowd-sourced label distributions. *ACL*, 2019.
>
> [7] Wu et al. Joint acne image grading and counting via label distribution learning. *ICCV*, 2019.
>
> [8] Li et al. Ambiguity-aware breast tumor cellularity estimation via self-ensemble label distribution learning. *Medical Image Analysis*, 2023.
>
> [9] Chen et al. Attention-guided discriminative region localization and label distribution learning for bone age assessment. *IEEE JBHI*, 2021.
>
> [10] Qin et al. Multitask deep label distribution learning for blood pressure prediction. *Information Fusion*, 2023.
>
> [11] Geng. Label distribution learning. *IEEE TKDE*, 2016.
>
> [12] Jia et al. LIMEFLDL: A local interpretable model-agnostic explanations approach for label distribution learning. *ICML*, 2025.
>
> [13] Wang and Geng. Theoretical analysis of label distribution learning. *AAAI*, 2019.
>
> [14] Xu, Liu, and Geng. Label enhancement for label distribution learning. *IEEE TKDE*, 2021.
>
> [15] Xu and Zhou. Incomplete label distribution learning. *IJCAI*, 2017.
>
> [16] Wang and Geng. Classification with label distribution learning. *IJCAI*, 2019.
>
> [17] Kou et al. Exploiting multi-label correlation in label distribution learning. *IJCAI*, 2024.
>
> [18] Kou et al. RankMatch: A novel approach to semi-supervised label distribution learning leveraging inter-label correlations. *NeurIPS*, 2025.

---

> > ### Comment · Reviewer_Jzzj · 2025-11-27
> >
> > 1. The authors themselves claim "applications of GLD are yet to be explored", my point is "If so, what is the point to propose a generalized label distribution as a solution to solve the drawbacks of label distribution".
> > 2. yes, "LD provides a nuanced framework to model label ambiguity/polysemy and captures richer information than traditional single-label or multi-label formulations", so LD is a kind of methods to solve the label ambiguity problem, you can develop more advanced LD-based methods to solve the label ambiguity problem. Thus, the motivation should come from solving label ambiguity problem instead of the drawbacks of LD.
> > 3. Pls refer to 2.
> >
> > To sum up, I still believe the motivation is not sound, and the contribution is very limited for ICLR.

---

> > > ### Author Response · Authors · 2025-11-28
> > >
> > > Thank you for your continued feedback.
> > >
> > > 1. Our intention was to express a forward-looking expectation for the community, not to undermine the value of the field nor to weaken the motivation of GLD. We will revise the wording accordingly to avoid any possible misunderstanding.
> > > 2. Developing more advanced LD-based methods *only treats the symptoms rather than the root cause*. Classical LD is constrained by a fixed probability-simplex representation, which imposes structural limitations that no algorithmic improvement can fundamentally overcome. This is what the paper consistently emphasizes. Our motivation for proposing GLD is precisely to remove these built-in representational constraints, enabling the field to grow beyond the narrow space defined by traditional LD. Without addressing this foundational limitation, continuing to build so-called "advanced LD methods" would only reinforce the same bottleneck and hinder long-term development.
> > >
> > > Thank you again for your response! We still hope the Reviewer will understand our motivation/contribution, and we will remain available for further clarification and look forward to the Reviewer’s reply.

---

> ### Author Response · Authors · 2025-11-26
>
> Dear Reviewer Jzzj,
>
> Hope this message finds you well.
>
> We would sincerely appreciate it if you could take a moment to provide feedback on our rebuttal, as the deadline for the discussion process is approaching. We believe that we have fully addressed your concerns, and your follow-up opinion is important.
>
> To briefly restate two key points clarified in the rebuttal:
>
> + GLD is compatible with the broad application spectrum of traditional LD, and it also enables potential applications beyond LD.
> + The purpose of proposing GLD is to better model label ambiguity/polysemy; pointing out the limitations of LDL supports this motivation rather than being misleading.
>
> We highly value your feedback and look forward to any further comments or suggestions you may have. Thank you for your time and valuable contribution to the review process!

---

### Official Review · Reviewer_Ca62 · 2025-10-30

**Soundness:** 2
**Presentation:** 3
**Contribution:** 3
**Rating:** 4
**Confidence:** 4

**Summary:**

This paper proposes the concept of a generalized label distribution (GLD) as a new representation of label ambiguity to address the inherent limitations of Label Distribution (LD) in supervised learning, which include inconsistency with raw data, disruption of inter-sample order, and inability to handle Out-of-Distribution (OOD) samples and negative label correlations. Moreover, inspired by the three types of algorithms for learning LDs (e.g., problem transformation, algorithm adaptation and specialized algorithm), this paper designs three Inspired by the distribution of learning labels, this paper designs three algorithms for learning GLDs: GLD-SVR corresponds to problem transformation, GLD-kNN corresponds to algorithm adaptation, and GLD-BFGS corresponds to specialized algorithm. They also adapt existing LDL methods (e.g., GLD-DF, GLD-LRR) to extend their applicability to GLD.

**Strengths:**

1. Through theoretical proofs and analyses in this paper, the author's definitions and proposals regarding GLD breaks through the limitations of LD, which were lacking in previous papers.
2. Theoretical analyses (mutual information, generalization bounds) are comprehensive.
3. The related work in this paper is comprehensive, citing classic LD literature and cross-domain studies like multi-label learning and OOD detection clearly distinguishing GLD from existing work. The structure of the paper is clear.

**Weaknesses:**

1. The experiments in the paper are thorough, but ablation experiments for core components of GLD-BFGS are absent, such as the Mahalanobis distance loss, so it is unclear whether the performance improvement comes from the GLD representation or algorithmic details.
2. All experiments are limited to low-dimensional labels and does not validate on high-dimensional tasks. GLD’s advantage persistence under label dimensionality scaling remains unconfirmed.
3. The LD OOD threshold is an empirical setting, and without cross-validation or comparison with other LD OOD methods, it can lead to unfair comparisons that exaggerate GLD's OOD advantages.
4. Key scales of the real datasets (e.g., the sample sizes, feature dimension for jaf/wq/enb) are not reported, and large-scale validation is not conducted, raising concerns about the generalizability of the results.
5. Robustness testing lacks quantitative descriptions of noise intensity, such as the standard deviation of Gaussian noise or the label zeroing rate, which reduces the reproducibility GLD's anti-interference results and weakens robustness claims.

**Questions:**

1. Could you add ablation experiments for GLD-BFGS’s core components? E.g.: Contrast GLD-BFGS with Mahalanobis loss vs. MSE loss to isolate loss function impact.
2. Could you validate GLD on high-dimensional label datasets?
3. Could you optimize the LD OOD threshold via cross-validation, or compare it with alternative LD OOD methods? This verifies if GLD’s OOD advantage is not due to LD metric bias.
4. Could you provide full scales of real datasets (sample size, feature dimension, label count) and validate GLD on large-scale datasets? This strengthens generalization assessments.
5. Could you supplement quantitative noise intensities in robustness tests?

---

> ### Author Response · Authors · 2025-11-16
> **Response to the comments of Reviewer Ca62 (Part I)**
>
> Many thanks for your comprehensive comments! We have provided point-by-point responses to your questions below.
>
> ---
>
> **1. Response about ablation experiments:** We appreciate this constructive suggestion. We conducted ablation experiments on GLD-BFGS. The table below records the performance differences between using the squared Mahalanobis loss ($\text{M}^2$) and the MSE loss, where **bold** values indicate performance improvements brought by $\text{M}^2$, and "-" indicates no significant change.
>
> |Datasets|Clark|$\mu_{\textbf{KLD}}$|Ham.|S. Acc.|Spear.|Ken.$^{\prime}$|OOD Err.|
> |-|-|-|-|-|-|-|-|
> |artf|-|-|-|-|-|-|-|
> |jura|**-.0054**|**0.74%**|-|-|**.0039**|**.0047**|-|
> |wq|.0098|-1.38%|**-.0002**|-0.06%|-.0111|-.0157|0.62%|
> |scm1d|**-.0002**|**0.06%**|**-.0011**|**0.35%**|-|**.0008**|**-0.38%**|
> |jaf|**-.0269**|**3.04%**|**-.0109**|**1.97%**|**.0196**|**.0120**|**-2.13%**|
> |atp1d|**-.0084**|**3.47%**|**-.0058**|**1.55%**|**.0152**|**.0115**|**-1.01%**|
> |enb|-|-|**-.0001**|**0.01%**|-.0003|-.0002|**-0.01%**|
>
> The results show that $\text{M}^2$ consistently provides equal or better performance across most datasets and evaluation metrics. The only exception is the **wq** dataset, which we attribute to the highly discrete nature of its relative-degree values.
>
> **2. Response about high-dimensional label datasets:** Unfortunately, no GLD datasets with high-dimensional label spaces are currently available. To address this limitation, we can construct such a dataset by extending Eqs. (59) - (62), where each element of the vector $\boldsymbol{w}$ is independently sampled from a uniform distribution over $[-0.5, 0.5]$. Based on this procedure, we generated an artificial dataset with a 50-dimensional label space and evaluated both LD-BFGS and GLD-BFGS on it. The results are summarized below.
>
> |Datasets|Clark|$\mu_{\textbf{KLD}}$|Ham.|S. Acc.|Spear.|Ken.$^{\prime}$|OOD Err.|
> |-|-|-|-|-|-|-|-|
> |LD-BFGS|3.422±.024|61.89%±.008|.2706±.006|00.00%±.000|**.9516**±.002|.8052±.004|79.55%±.143|
> |GLD-BFGS|**2.130**±.110|**76.73%**±.273|**.0527**±.006|**07.00%**±.035|.9509±.023|**.8726**±.009|**71.96%**±.048|
>
> It is worth noting that, due to the uniformly generated data and the high-dimensional label space, metrics of classification/multi-label performance no longer carry clear interpretive value.

---

> ### Author Response · Authors · 2025-11-16
> **Response to the comments of Reviewer Ca62 (Part II)**
>
> (Cont'd)
>
> **3. Response about LD OOD threshold:** We would like to clarify that the LD OOD threshold is *not* a parameter that can be freely optimized, since setting it to $\infty$ would trivially classify all OOD samples as correct. In [1], $\delta_0$ is introduced to represent a theoretically near-random performance level, meaning that evaluating with a threshold larger than $\delta_0$ offers little meaningful reference. The value $\delta_0 / 2$ is provided merely for illustrative purposes.
>
> To demonstrate this, we use the **jura** dataset as an example and report OOD Err. of LD-BFGS w.r.t. varying threshold values:
>
> |Threshold|0.05|0.1|0.1612 ($\delta_0 / 2$)|0.2|0.3|0.3224 ($\delta_0$)|
> |-|-|-|-|-|-|-|
> |**OOD Err.**|60.69%|60.69%|57.12%|54.33%|36.37%|32.44%|
>
> As shown, even when the threshold is relaxed to $\delta_0$, the OOD error of LD-BFGS remains 32.44%, which is still significantly higher than 15.43% achieved by GLD-BFGS. This confirms that the observed OOD advantage of GLD is not due to a bias in the LD metric.
>
> **4. Response about large-scale datasets:** We thank the Reviewer for this suggestion. The table below provides the full scales of the real datasets used in our experiments.
>
> |Datasets|#Samples|#Features|#Labels|
> |-|-|-|-|
> |artf|200|3|3|
> |jaf|213|243|6|
> |atp1d|337|411|6|
> |jura|359|15|3|
> |enb|768|8|2|
> |wq|1054|16|14|
> |scm1d|9803|280|16|
>
> As shown, our experiments already include large-scale datasets, such as scm1d with 9803 samples. In addition, we evaluate GLD on the **jaf** dataset with real images (256 $\times$ 256 facial images) to demonstrate its capability to handle high-dimensional features.
>
> Experimental settings:
>
> + Architecture: ResNet-50 backbone.
> + Training: 10 independent runs, SGD optimizer (learning rate 2e-3, 200 epochs).
>
> Results:
>
> |Datasets|Clark|$\mu_{\textbf{KLD}}$|Ham.|S. Acc.|Spear.|Ken.$^{\prime}$|OOD Err.|
> |-|-|-|-|-|-|-|-|
> |Tabular|.6781±.054|47.83%±.091|.2134±.039|26.74%±.092|.5116±.096|.4094±.069|45.51%±.116|
> |Image|**.4674**±.049|**74.60%**±.049|**.1136**±.023|**51.82%**±.094|**.7051**±.057|**.5857**±.056|**24.55%**±.084|
>
> **5. Response about robustness tests:** In our robustness tests, we supplemented quantitative noise intensities as follows:
>
> 1. Gaussian noise: Each description degree is perturbed by Gaussian noise with mean 0 and standard deviation 0.1.
> 2. Random missing: A proportion of 20% of description degrees are randomly set to 0 to simulate missing data.
> 3. Random emphasis: A proportion of 50% of description degrees are randomly set to 1 to simulate exaggerated data.
>
> ---
>
> In summary, we especially appreciate the Reviewer’s suggestion regarding high-dimensional label datasets, as discussions on this topic remain scarce in the field of label distribution learning. Please let us know if our responses address your concerns, and we welcome any further questions or discussion!
>
> [1] Li et al. Approximately Correct Label Distribution Learning. *ICML*, 2025.

---

> > ### Comment · Reviewer_Ca62 · 2025-11-20
> > **l am retaining my original score**
> >
> > While l appreciate the authors' efforts to address the concerns raised, the experimental results included in the rebuttal lack the rigor needed to justify a higher rating. Accordingly, l am retaining my original score.

---

> > > ### Author Response · Authors · 2025-11-21
> > >
> > > While we respect the Reviewer's decision, we disagree with the assessment that the results "lack rigor", especially given that *the basis for this critique was not clearly specified*.
> > >
> > > We would like to clarify that the experimental results provided in the rebuttal were conducted following standard, widely accepted evaluation protocols in the LDL community, and were intended to directly address the Reviewer's earlier concerns.

---

### Official Review · Reviewer_Mfx3 · 2025-10-31

**Soundness:** 4
**Presentation:** 4
**Contribution:** 4
**Rating:** 6
**Confidence:** 4

**Summary:**

This paper revisits the problem of representing label ambiguity and theoretically demonstrates that the widely adopted label distribution (LD) representation exhibits several intrinsic limitations. To address these issues, the authors propose a novel representation, termed generalized label distribution (GLD), which unifies various existing forms of label representations under a single framework. GLD is theoretically analyzed and supported by newly designed learning algorithms, whose effectiveness is demonstrated through both analytical and experimental evaluations.

**Strengths:**

1. This paper provides a creative and interesting perspective: its novelty lies in the theoretical deconstruction of the conventional LD paradigm, which reveals inherent inconsistencies and motivates the formulation of a more expressive and principled GLD framework with broad potential impact.
2. This paper provides rigorous and convincing theoretical analyses that comprehensively expose the limitations of LD and justify the superiority of GLD and its associated learning algorithms.
3. The proposed GLD exhibits a strong unification property, elegantly and generally recovering raw data without information loss, and deriving other label forms.
4. The empirical evaluation of GLD algorithms reinforces the theoretical claims and enhances the overall credibility of the proposed framework.

**Weaknesses:**

1. Some mathematical descriptions are redundant and contain inaccuracies or lack sufficient elaboration. Please refer to the specific questions below.
2. The limitations of GLD are not thoroughly discussed. A more explicit elaboration on these aspects would better clarify the respective scenarios where LD and GLD are most applicable.

**Questions:**

1. Theorems 3.6-3.9 lack adequate explanation in the manuscript. What key insights or conclusions do each of these theorems reveal?
2. There appear to be potential issues with some mathematical notations: Is $i$ in Eqn. (2) redundant? Should $L_j$ in Lines 104-105 be denoted as $L_j^{*}$ instead?
3. While GLD is presented as a powerful framework, it may be challenging to apply in scenarios with limited raw data due to human or time constraints. What are the specific requirements for constructing GLD, and are there strategies to handle data scarcity?
4. The limitations of GLD are not thoroughly discussed. Are there specific aspects or scenarios where traditional LD remain irreplaceable? Could the authors provide a more in-depth discussion of its potential limitations?

---

> ### Author Response · Authors · 2025-11-16
> **Response to the comments of Reviewer Mfx3**
>
> Many thanks for your precious comments! Responses to your concerns are as follows.
>
> ---
>
> **1. Response about mathematical notations:** We thank the Reviewer for pointing this out. The suggested modifications are indeed correct, and we will revise them accordingly.
>
> **2. Response about Theorems 3.6-3.9:** We provide the following key insights and conclusions, grouped by their focus.
>
> + **A. Variance analysis of GLD-$k$NN predictions:**
>   + Theorem 3.6: This theorem formalizes that the variance of predictions from GLD-$k$NN is no larger than that of conventional LD-$k$NN. In other words, GLD provides more stable predictions while retaining unbiasedness. We can use GLD to reduce uncertainty in LD estimation.
>   + Theorem 3.7: This theorem is similar to Theorem 3.6, but it discusses logical label prediction and has certain conditions.
> + **B. Generalization performance of specialized GLD algorithms:**
>   + Theorem 3.8: This theorem provides an upper bound on the empirical Rademacher complexity of the squared Mahalanobis distance loss function used in specialized GLD model like GLD-BFGS, laying the groundwork for the proof of Theorem 3.9.
>   + Theorem 3.9: This theorem gives a generalization bound for specialized GLD algorithms. It highlights that GLD can generalize reliably to unseen data.
>
> **3. Response about requirements for constructing GLD:** We thank the reviewer for the valuable comment. Considering that the LD is derived from specific raw crowd-sourced data, an important observation is that most label distribution learning (LDL) tasks and multi-target regression (MTR) problems can be naturally cast as GLD problem families, and, in essence, *the prerequisites for constructing datasets in these tasks are consistent*. The applicability of GLD is quite flexible: as long as the label space exhibits discrete semantics, a problem can naturally be modeled under the GLD framework. In our experiments, the datasets we selected are all classical LDL/MTR benchmarks, and after being converted into GLD tasks, they remain meaningful and practically relevant.
>
> **4. Response about strategies to handle data scarcity:** GLD is essentially a reformulation that expresses label ambiguity/polysemy in a more accurate form. This means that existing strategies developed for LDL to address data scarcity (e.g., low-rank recovery [1], data augmentation via oversampling [2], and few-shot domain adaptation [3]) can be readily adapted and applied within the GLD framework.
>
> **5. Response about limitations:** We thank the reviewer for raising this important point. We have carefully discussed the limitations of GLD:
>
> + **Fixed raw data volume:** When the raw data for each sample is of a fixed total amount, GLD theoretically does not offer an advantage over traditional LD. For example, **Theorem 3.3** and **Theorem 3.6** in our paper explicitly support this observation.
> + **Continuous label semantics:** When the label space exhibits continuous semantics, GLD lacks a clear and interpretable meaning. In such cases, traditional LD remains irreplaceable for simulating this uncertainty.
>
> ---
>
> In summary, we especially thank the reviewer for pointing out the notational redundancy and for guiding us to reflect more deeply on the limitations of our framework. Please let us know if our responses address your concerns, and we welcome any further questions or discussion!
>
> [1] Xu and Zhou. Incomplete label distribution learning. *IJCAI*, 2017.
>
> [2] González et al. Synthetic sample generation for label distribution learning. *Information Sciences*, 2021.
>
> [3] Wu, Li, and Jia. Domain adaptation for label distribution learning. *IEEE TBD*, 2024.

---

> ### Author Response · Authors · 2025-11-26
>
> Dear Reviewer Mfx3,
>
> Hope this message finds you well.
>
> We kindly hope you could take a moment to provide feedback on our rebuttal. We have carefully addressed all raised concerns with additional theoretical explanations and clarifications of the limitations, and we believe these directly resolve the key questions identified in the initial review.
>
> Your follow-up opinion is important, and we would be truly grateful if you could kindly share your updated thoughts. Thank you for your time and valuable contribution to the review process!

---

### Official Review · Reviewer_bNBM · 2025-11-01

**Soundness:** 3
**Presentation:** 3
**Contribution:** 2
**Rating:** 6
**Confidence:** 2

**Summary:**

The paper argues that conventional Label Distribution (LD) is intrinsically limited-it can be misaligned with raw targets, distort inter-sample order, and struggles with OOD detection and negative label correlations.

It proposes a Generalized Label Distribution (GLD): a per-label bijective affine map of raw targets into $[-1,1]^c$ that is invertible and can be losslessly converted into other label forms (LL, TL, LR, SL).

Experiments on synthetic and real datasets evaluate LL/LD prediction, ranking, and an OOD metric; GLD variants often outperform LD counterparts and show robustness under several noise processes.

**Strengths:**

1. GLD is bijective to raw targets, preserves inter-sample order, and permits lossless conversions among multiple label forms.

2. Clear information comparison (mutual-information inequality) and generalization bounds under a Mahalanobis loss.

3. Broad metrics (LL/LD prediction, rank correlations, an OOD metric) across synthetic and real datasets; GLD methods frequently top LD baselines.

**Weaknesses:**

1. Many LDL settings begin with label distributions collected from annotators; no continuous $q$ exists. The paper should clarify which problem families naturally expose $q$, and when GLD is more than renormalized multi-target regression.

2. The Mahalanobis loss relies on $\Sigma$; details on estimating/regularizing $\Sigma$ and sensitivity are light.

3. Several benchmarks are classic multi-output regression. To convince the LDL community, include canonical LDL datasets with crowd distributions and compare to Dirichlet/soft-label baselines and modern LDL methods.

**Questions:**

1. For OOD, please compare against standard embedding-space OOD baselines and report AUROC/FPR95, not only a bespoke OOD-Err.

2. Can you add ablations indicating that gains are not from the Mahalanobis loss alone (e.g., swap it into LD models)?

---

> ### Author Response · Authors · 2025-11-16
> **Response to the comments of Reviewer bNBM (Part I)**
>
> Many thanks for your comprehensive comments! We have provided point-by-point responses to your questions below.
>
> ---
>
> **Comment 1:** Many LDL settings begin with LDs collected from annotators; no continuous $q$ exists. The paper should clarify which problem families naturally expose the raw data $q$, and when GLD is more than renormalized multi-target regression (MTR).
>
> **Response:** We would like to clarify that in most LDL settings, annotators do *not* provide LDs directly, since this would incur a high human time cost. Instead, the LD is derived from the raw crowd-sourced data, which corresponds to $q$ in our formulation. The applicability of GLD is quite flexible: *as long as the label space exhibits discrete semantics, a problem can naturally be modeled under the GLD framework*. Consequently, most LDL tasks and MTR problems can be naturally cast as GLD problem families.
>
> GLD goes beyond renormalized MTR in two key aspects: 1) it unifies other forms of label-polysemy data, as illustrated in **Table 1**, allowing diverse label representations to be modeled within a single framework; 2) it provides direct and interpretable correlation information among labels, thanks to the well-defined data range of $[-1,\, 1]_{\mathbb{R}}^c$ introduced in **Definition 3.2**. These capabilities are *not* achievable within the MTR framework. At the same time, GLD can still recover the original MTR data (which corresponds to $q$), demonstrating that it generalizes MTR while preserving backward compatibility.
>
> **Comment 2:** The Mahalanobis loss relies on $\boldsymbol{\Sigma}$; details on estimating/regularizing $\boldsymbol{\Sigma}$ and sensitivity are light.
>
> **Response:** We thank the Reviewer for pointing this out. Briefly, we use a Ledoit-Wolf method to estimate $\boldsymbol{\Sigma}$, and $\alpha$ gradually transitions from identity regularization to the estimated covariance inverse to ensure stability. The corresponding code is as follows:
>
> ```python
> tmp = float(current_iteration) / max_iterations
> if tmp >= .5 and current_iteration % 100 == 0:
>     from sklearn.covariance import LedoitWolf
>     alpha = (tmp - .5) * 2
>     eps = G - G_pred
>     cov = LedoitWolf().fit(eps).covariance_
>     sigma_inv = alpha * np.linalg.inv(cov) + (1 - alpha) * np.eye(n_outputs)
> ```
>
> We also commit that, if the paper is accepted, we will release the source code. All implementation details will be fully accessible.
>
> **Comment 3:** Several benchmarks are classic multi-output regression. To convince the LDL community, include canonical LDL datasets with crowd distributions and compare to Dirichlet/soft-label baselines and modern LDL methods.
>
> **Response:** For LDL datasets with *continuous label-space semantics*, such as crowd distributions, the advantage of GLD is not immediately clear, which we also note as a limitation in the conclusion of our manuscript. For LDL datasets with *discrete label-space semantics*, some raw data are difficult to access, and the methods used for renormalization are often unclear. Despite these challenges, we have reconstructed three datasets previously used in LDL studies, **jaf**, **jura**, and **wq**, and additionally created a synthetic dataset **artf** to provide more evidence and convince the LDL community of GLD’s utility.
>
> Regarding baselines, Dirichlet/soft-label methods rely on strong assumptions and are not considered mainstream LDL solutions. We have adapted some modern LDL methods for fair comparison (LD-DF, LD-LRR, and LD-Delta), as described in **Section 4.1**.

---

> ### Author Response · Authors · 2025-11-16
> **Response to the comments of Reviewer bNBM (Part II)**
>
> (Cont'd)
>
> **Comment 4:** For OOD, please compare against standard embedding-space OOD baselines and report AUROC/FPR@95, not only a bespoke OOD-Err.
>
> **Response:** We appreciate the Reviewer’s suggestion. However, a bespoke OOD measure is necessary in our case, because there is currently *no* established method to directly convert LD predictions into OOD classification decisions. Therefore, following the conditions in Eq. (69), we customized corresponding AUROC and FPR@95 metrics for our setting. The results are summarized in the table below.
>
> |Datasets|Algorithms|AUROC|FPR@95|
> |-|-|-|-|
> |artf|LD-BFGS|.633±.193|.752±.307|
> ||GLD-BFGS|**.977**±.021|**.044**±.040|
> |jura|LD-BFGS|.495±.033|.950±.003|
> ||GLD-BFGS|**.721**±.076|**.774**±.134|
> |wq|LD-BFGS|.504±.052|.949±.007|
> ||GLD-BFGS|**.591**±.058|**.831**±.111|
> |scm1d|LD-BFGS|.554±.020|.942±.003|
> ||GLD-BFGS|**.870**±.019|**.791**±.032|
> |jaf|LD-BFGS|**.647**±.181|**.727**±.295|
> ||GLD-BFGS|.557±.161|.876±.209|
> |atp1d|LD-BFGS|.735±.058|.896±.021|
> ||GLD-BFGS|**.937**±.068|**.141**±.178|
> |enb|LD-BFGS|.146±.046|.992±.004|
> ||GLD-BFGS|**.958**±.021|**.273**±.226|
>
> As shown, GLD still demonstrates a clear advantage in terms of pure OOD classification performance.
>
> **Comment 5:** Can you add ablations indicating that gains are not from the Mahalanobis loss alone (e.g., swap it into LD models)?
>
> **Response:** Specifically, in **Tables 2 & 4**, every two consecutive rows can be interpreted as swapping GLD models into LD models: the *white-background rows* correspond to the original LD methods, while the *gray-background rows* represent our GLD variants.
>
> Furthermore, we conducted ablation experiments on GLD-BFGS. The table below records the performance differences between using the squared Mahalanobis loss ($\text{M}^2$) and the MSE loss, where **bold** values indicate performance improvements brought by $\text{M}^2$, and "-" indicates no significant change.
>
> |Datasets|Clark|$\mu_{\textbf{KLD}}$|Ham.|S. Acc.|Spear.|Ken.$^{\prime}$|OOD Err.|
> |-|-|-|-|-|-|-|-|
> |artf|-|-|-|-|-|-|-|
> |jura|**-.0054**|**0.74%**|-|-|**.0039**|**.0047**|-|
> |wq|.0098|-1.38%|**-.0002**|-0.06%|-.0111|-.0157|0.62%|
> |scm1d|**-.0002**|**0.06%**|**-.0011**|**0.35%**|-|**.0008**|**-0.38%**|
> |jaf|**-.0269**|**3.04%**|**-.0109**|**1.97%**|**.0196**|**.0120**|**-2.13%**|
> |atp1d|**-.0084**|**3.47%**|**-.0058**|**1.55%**|**.0152**|**.0115**|**-1.01%**|
> |enb|-|-|**-.0001**|**0.01%**|-.0003|-.0002|**-0.01%**|
>
> The results show that $\text{M}^2$ consistently provides equal or better performance across most datasets and evaluation metrics. The only exception is the **wq** dataset, which we attribute to the highly discrete nature of its relative-degree values.
>
> ---
>
> We particularly appreciate the Reviewer’s suggestion to report AUROC and FPR@95, as this makes our work more accessible and interpretable to researchers in the OOD community. Please let us know if our responses above address your concerns, and we welcome any further questions or discussions!

---

> > ### Comment · Reviewer_bNBM · 2025-11-26
> > **Retain my original score**
> >
> > I thank the authors for the detailed rebuttal. After careful consideration, I have decided to retain my original score.

---

> ### Author Response · Authors · 2025-11-26
>
> Dear Reviewer bNBM,
>
> Hope this message finds you well.
>
> We sincerely hope you could take a moment to provide feedback on our rebuttal, as the deadline for the discussion process is approaching. We have carefully addressed all raised concerns with additional experiments and clarifications, and we believe the new results directly resolve the key questions.
>
> Your follow-up opinion is important, and we highly value your feedback and look forward to any further comments or suggestions you may have. Thank you for your time and valuable contribution to the review process!

---

### Meta-Review · Area_Chair_7nSM · 2026-01-05

**Summary:**

This paper proposes Generalized Label Distribution (GLD) as a unified representation to model label ambiguity, aiming to overcome intrinsic limitations of conventional LD. This work also provides a systematic theoretical analysis, motivates several GLD learning algorithms, and presents extensive experiments across synthetic and real datasets.

The reviews are mixed. Two reviewers view the work as technically strong and conceptually meaningful, highlighting the theoretical depth and the unifying nature of GLD. In contrast, one reviewer raises concerns regarding empirical results/analysis, and another reviewer strongly questions whether GLD addresses a meaningful problem beyond LD. The rebuttal is detailed and technically thorough, addressing most concrete experimental and theoretical concerns, but it does not fully change the opinions of the more skeptical reviewers.

**Reviewer Concerns:**

Two major concerns remain after the rebuttal.

1. Reviewer Ca62 raises concerns regarding the empirical results and analysis. While the authors provide additional results in the rebuttal, the reviewer remains unconvinced.
2. Reviewer Jzzj raises concerns regarding the motivation of this GLD.

After reading the paper myself, I partially agree with the concerns. In particular, I note that the advantages of GLD are more clearly demonstrated on synthetic datasets (e.g., Table 2), where the representational assumptions are explicitly controlled. On real-world datasets, however, the improvements over LD baselines are generally less pronounced. To some extent, this empirical pattern aligns with and partially supports the concerns raised by the reviewers regarding the practical impact of GLD.

**Reviewer Scores:**

Reviewer bNBM: Rated the paper slightly above the acceptance threshold and retained the original score after rebuttal. Given the additional experiments and clarifications, this score appears stable, with no clear indication of an upward change.

Reviewer Mfx3: Gave a positive assessment on soundness, presentation, and contribution, and rated the paper above the threshold. Although no explicit post-rebuttal update was provided, the rebuttal adequately addressed the raised concerns, and the score would likely remain unchanged.

Reviewer Ca62: Rated the paper slightly below the acceptance threshold and retained the original score, citing insufficient rigor despite additional experiments. Based on the comments, a score change is unlikely.

Reviewer Jzzj: Assigned a strong reject and maintained this position throughout the discussion, remaining unconvinced by the rebuttal.

---

### Decision · Program_Chairs · 2026-01-26

Reject